# Improving Deep Learning Interpretability by Saliency Guided Training

**Aya Abdelsalam Ismail,  Soheil Feizi** *,  **Héctor Corrada Bravo** *

{asalam,sfeizi}@cs.umd.edu,   corradah@gene.com
Department of Computer Science, University of Maryland
Data Science and Statistical Computing, Genentech, Inc.

## Abstract

Saliency methods have been widely used to highlight important input features in model predictions. Most existing methods use backpropagation on a modified gradient function to generate saliency maps. Thus, noisy gradients can result in unfaithful feature attributions. In this paper, we tackle this issue and introduce a *saliency guided training*[†] procedure for neural networks to reduce noisy gradients used in predictions while retaining the predictive performance of the model. Our saliency guided training procedure iteratively masks features with small and potentially noisy gradients while maximizing the similarity of model outputs for both masked and unmasked inputs. We apply the saliency guided training procedure to various synthetic and real data sets from computer vision, natural language processing, and time series across diverse neural architectures, including Recurrent Neural Networks, Convolutional Networks, and Transformers. Through qualitative and quantitative evaluations, we show that saliency guided training procedure significantly improves model interpretability across various domains while preserving its predictive performance.

## 1   Introduction

Deep Neural Networks (DNNs) have been widely used in a variety of different tasks [31, 26, 43, 37]; yet interpreting complex networks remains a challenge. Reliable explanations are necessary for critical domains like medicine, neuroscience, finance, and autonomous driving [9, 34]. Explanations are also useful for model debugging [63, 35]. As a result, various interpretability methods were developed to understand DNNs [5, 49, 22, 53, 52, 32, 51]. A common approach for understanding model decisions is to identify features in the input that highly influenced the final classification decision [6, 63, 53, 52, 47, 36, 64]. Such approaches, known as *saliency maps*, often use gradient calculations to assign an *importance* score to individual features, reflecting their influences on the model prediction.

Saliency methods aim to highlight meaningful input features in model predictions to humans; however, the maps produced are often noisy (i.e., contain visual noise). To improve the faithfulness of saliency maps, explanations methods that depend on more than one or higher-order gradient calculations were developed. For example, SmoothGrad [52] reduces saliency noise by adding noise to the input multiple times and then taking the average of the resulting saliency maps for each input. Integrated gradients [53], DeepLIFT [47] and Layer-wise Relevance Propagation [5] backpropagate through a modified gradient function [3] while Singla et al. [51] studies the use of higher-order gradients in saliency maps.

---

*Authors contributed equally
[†]Code: https://github.com/ayaabdelsalam91/saliency_guided_training

35th Conference on Neural Information Processing Systems (NeurIPS 2021).

In this paper, we take a different approach to improve the interpretability of deep neural networks—instead of developing yet another saliency method, we propose a new *training procedure* that naturally leads to improved model explanations using current saliency methods. Our proposed training procedure, called *saliency guided training*, trains models that produce sparse, meaningful, and less noisy gradients without degrading model performance. This is done by iteratively masking input features with low gradient values (i.e., less important features) and then minimizing a loss function that combines (a) the KL divergence [27] between model outputs from the original and masked inputs, and (b) the appropriate loss function for the model prediction. This procedure reduces noise in model gradients without sacrificing its predictive performance.

To demonstrate the effectiveness of our proposed saliency guided training approach, we consider a variety of classification tasks for images, language, and multivariate time series across diverse neural architectures, including Convolutional Neural Networks (CNNs), Recurrent Neural Network (RNNs), and Transformers. In particular, we observe that using saliency guided training in image classification tasks leads to a reduction in visual saliency noise and sparser saliency maps, as shown in Figure 2. Saliency guided training also improves the comprehensiveness of the produced explanations for sentiment analysis, and fact extraction tasks as shown in Table 1.

In multivariate time series classification tasks, we observe an increase in the precision and recall of saliency maps when applying the proposed saliency guided training. Interestingly, we also find that the saliency guided training reduces the vanishing saliency issue of RNNs [20] as shown in Figure 6. Finally, we note that although we use the vanilla gradient for masking in the saliency guided training procedure, we observe significant improvements in the explanations produced after training by several other gradient-based saliency methods.

## 2 Background and Related Work

Interpretability is a rapidly growing area with several diverse lines of research. One strand of interpretability considers post-hoc explanation methods, aiming to explain why a trained model made a specific prediction for a given input. Post-hoc explanation methods can be divided into gradient-based methods [6, 53, 52, 47, 36, 46] that can be reformulated as computing backpropagation for a modified gradient function and perturbation-based approaches [63, 54, 42, 57] that perturb areas of the input and measure how much this changes the model output. Perkins et al. [40] uses gradients for feature selection through Grafting. Another line of works aims to measure the reliability of interpretability methods. This can be done by creating standardize benchmarks with interpretability metrics [18, 21, 11, 56, 45, 41] or debugging explanations [1, 24, 15, 2] by identifying test cases where explanations fail. Others [4, 12, 44, 61, 20] focus on modifying neural architectures for better interpretability. Similar to our line of work, Ghaeini et al. [14] and Ross et al. [44] incorporate explanations into the learning process. However, Ghaeini et al. [14] relies on the existence of the ground truth explanations while Ross et al. [44] relies on the availability of annotations about incorrect explanations for a particular input. Our proposed learning approach does not rely on such annotations; since most datasets only have ground truth labels, it may not be practical to assume the availability of positive or negative explanations.

Input level perturbation during training has been previously explored. [33, 19, 60, 50] use attention maps to improve segmentation for weakly supervised localization. Wang et al. [59] incorporates attention maps into training to improve classification accuracy. DeVries and Taylor [10] masks out square regions of input during training as a regularization technique to improve the robustness and overall performance of convolutional neural networks. Our work focuses on a different task which is increasing model interpretability through training in a self-supervised manner.

In this paper, we evaluate our learning procedure with the following saliency methods: **Gradient (GRAD)** [6] is the gradient of the output w.r.t the input. **Integrated Gradients (IG)** [53] calculates a path integral of the model gradient to the input from a non-informative reference point. **DeepLIFT (DL)** [47] compares the activation of each neuron to a reference activation; the relevance is the difference between the two activations. **SmoothGrad (SG)** [52] samples similar input by adding noise to the input and then takes the average of the resulting sensitivity maps for each sample. **Gradient SHAP (GS)** [36] adds noise to the input, then selects a point along the path between a reference point and input, and computes the gradient of outputs w.r.t those points.

We demonstrate the effectiveness of our training procedure using several neural network architectures: Convolution neural networks (CNNs) including VGG-16 [48], ResNet [16] and Temporal Convolu-

tional Network (TCN) [38, 28, 7], a CNN that handles sequences; Recurrent neural networks (RNNs) including LSTM [17] and LSTM with Input-Cell Attention [20]; as well as Transformers [58].

## 3   Notation

First, consider a classification problem on the input data $\{(X_i, y_i)\}_{i=1}^n$ such that each $X = [x_1, \ldots, x_N] \in \mathbb{R}^N$ has $N$ features and $y$ is the label. Let $f_\theta$ denote a neural network parameterized by $\theta$. The standard training of the network involves minimizing the cross-entropy loss $\mathcal{L}$ over the training set as follows:

$$\underset{\theta}{\text{minimize}} \quad \frac{1}{n} \sum_{i=1}^n \mathcal{L}\left(f_\theta\left(X_i\right), y_i\right) \tag{1}$$

The gradient of the network output $f_\theta\left(X\right)$ with respect to the input $X$ is given by $\nabla_X f_\theta\left(X\right)$. Let $S(.)$ be a sorting function such that $S_e(Z)$ is the $e^{th}$ smallest element in $Z$. Hence, $S\left(\nabla_X f_\theta\left(X\right)\right)$ is the sorted gradient. We define the input mask function $M_k(.)$ such that $M_k(S(X), X)$ replaces all $x_i$ where $S(x_i) \in \{S_e\left(x_i\right)\}_{e=0}^k$ with a mask distribution, i.e., $M_k(S(X), X)$ removes the $k$ lowest features from $X$ based on the order provided by $S(X)$.

For a language input, we use $X = [x_1, \ldots, x_N]$ where $x_i \in \mathbb{R}^d$ is the feature embedding representing the $i^{th}$ word of the input. In that case, $S(X)$ would sort elements of $X$ based on the sum of the gradient of the embeddings for each word $x$ and $M_k(S(X), X)$ would mask the bottom $k$ words according to that sorting. For a multivariate time series input, we use $X = [x_{1,1}, \ldots, x_{F,1}, \ldots, x_{F,T}] \in \mathbb{R}^{F \times T}$ where $T$ is the number of time steps and $F$ is the number of features per time step. $x_{i,t}$ is the input feature $i$ at time $t$; sorting and masking would be done at the $x_{i,t}$ level.

For two discrete probability distributions $P$ and $Q$ defined on the same probability space $\mathcal{X}$, the Kullback–Leibler (KL) divergence [27] (or, relative entropy) from $Q$ to $P$ is given as $D_{\text{KL}}$:

$$D_{\text{KL}}(P \parallel Q) = \sum_{x \in \mathcal{X}} P(x) \log\left(\frac{P(x)}{Q(x)}\right). \tag{2}$$

## 4   Saliency Guided Training

Existing gradient-based methods can produce noisy saliency maps as shown in Figure 1. The saliency map noise may be partially due to some uninformative local variations in partial derivatives. Using a standard training procedure based on ERM (expectation risk minimization), the gradient of the model w.r.t. the input (i.e., $\nabla_X f_\theta\left(X\right)$) may fluctuate sharply via small input perturbations [52].

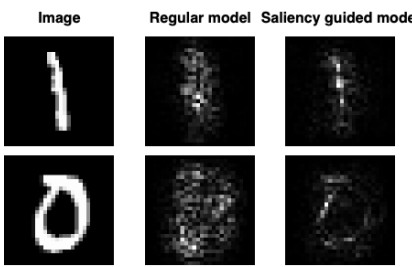

Figure 1: Saliency maps produced by typical training versus saliency guided training.

If gradient-based explanation methods faithfully interpret the model's predictions, irrelevant features should have gradient values close to zero. Building on this intuition, we introduce *saliency guided training*, a procedure to train neural networks such that input gradients computed from trained models provide more faithful measures to downstream (gradient-based) saliency methods. Saliency guided training aims to reduce gradient values of irrelevant features without sacrificing the model performance. During saliency guided training, for every input $X$, we create a new input $\widetilde{X}$ by masking the features with low gradient values as follows:

$$\widetilde{X} = M_k(S(\nabla_X f_\theta\left(X\right)), X) \tag{3}$$

$\widetilde{X}$ is then passed through the network which results in an output $f_\theta(\widetilde{X})$. In addition to the classification loss, the saliency guided training minimizes the KL divergence between $f_\theta(X)$ and $f_\theta(\widetilde{X})$ to ensure that the trained model produces similar output probability distributions over labels for both masked and unmasked inputs. The optimization problem for the saliency guided training is:

$$\underset{\theta}{\text{minimize}} \ \frac{1}{n} \sum_{i=1}^{n} \left[ \mathcal{L}\Big(f_\theta(X_i), y_i\Big) + \lambda D_{KL}\Big(f_\theta(X_i) \parallel f_\theta(\widetilde{X_i})\Big) \right] \tag{4}$$

where $\lambda$ is a hyperparameter to balance between the cross-entropy classification loss and the KL divergence term. Since this loss function is differentiable with respect to $\theta$, it can be optimized using existing gradient-based optimization methods. The KL divergence term encourages the model to produce similar outputs for the original input $X$ and masked input $\widetilde{X}$. For this to happen, the model will need to learn to assign low gradient values to irrelevant features in model predictions. This potentially results in sparse and more faithful gradients as shown in Figure 1.

**Masking functions:** In images and time series data, features with low gradients are replaced with random values within the feature range. In language tasks, the masking function replaces the low salient word with the previous high salient word. This allows us to emphasize on high salient words and remove non-salient ones while maintaining the sentence length. The selection of $k$ is dataset-dependent. It depends on the amount of irrelevant information in a training sample. For example, since most pixels in MNIST are uninformative, a larger $k$ is desired. Detailed hyperparameters used is available in the appendix. Note that, only input features are masked during the saliency guided training.

**Limitations:** (a) Compared to traditional training, our proposed training procedure is more computationally expensive. Specifically, the memory needed is doubled since now in addition to storing the batch, we are storing the masked batch as well. Similar to adversarial training, this training process is slow and takes a larger number of epochs to converge. For example, the standard training of a CIFAR-10 model usually takes on average 118 epochs to converge where each epoch is roughly 24 seconds. Using the saliency guided training, the convergence takes about 124 epochs where each epoch takes roughly 75 seconds (all experiments on the same GPU). (b) Our training procedure requires two hyperparameters $k$ and $\lambda$ which might require a hyperparameter search (we find that $\lambda = 1$ works well in all of our experiments).

---

**Algorithm 1:** Saliency Guided Training

---

**Given:** Training samples $X$, # of features to be masked $k$, learning rate $\tau$, hyperparameter $\lambda$

1 Initialize $f_\theta$

2 **for** $i \leftarrow 1$ **to** *epochs* **do**

3     **for** *minibatch* **do**

4         **Compute the masked input:**

            Get sorted index $I$ for the gradient of output with respect to the input.

            $I = S\Big(\nabla_X f_{\theta_i}(X)\Big)$

            Mask bottom $k$ features of the original input.

            $\widetilde{X} = M_k(I, X)$

5         **Compute the loss function:**

            $L_i = \mathcal{L}\Big(f_{\theta_i}(X), y\Big) + \lambda D_{KL}\Big(f_{\theta_i}(X) \parallel f_{\theta_i}(\widetilde{X})\Big)$

6         **Use the gradient to update network parameters:**

            $f_{\theta_{i+1}} = f_{\theta_i} - \tau \nabla_{\theta_i} L_i$

7     **end**

8 **end**

---

# 5 Experiments

All experiments have been repeated 5 times; the results reported below are the average of the 5 runs. Hyperparameters used for each experiment, along with the standard error bars and details on computational resources are available in supplementary materials.

## 5.1 Saliency Guided Training for Images

In the following section, we compare gradient-based explanations produced by regular training versus saliency guided training for MNIST [30] trained on a simple CNN [29], for CIFAR10 [25] trained on ResNet18 [16] and for BIRD [13] trained on VGG-16 [48]. Further details about the datasets and models are available in the supplementary material.

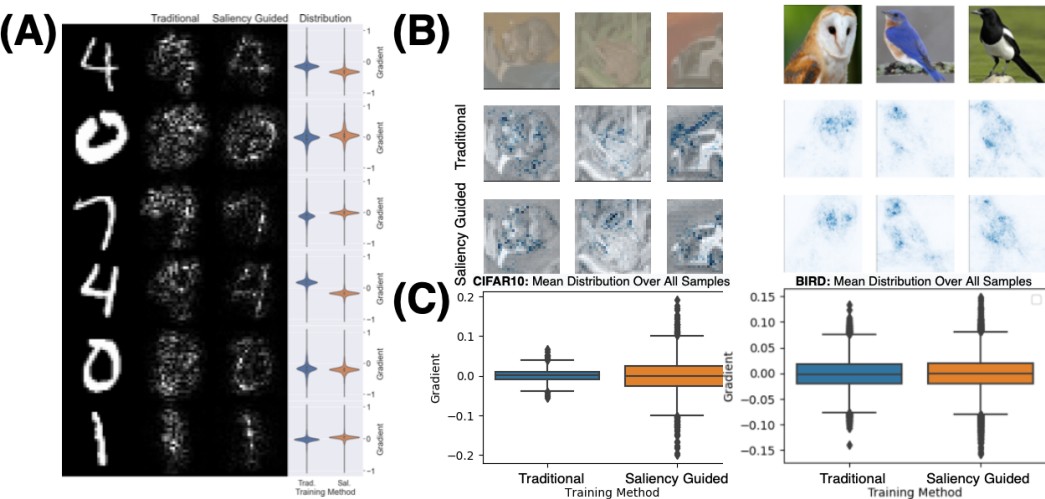

Figure 2: (A) Comparison between different training methods on MNIST along with distributions of gradient values in each sample. (B) Saliency maps for CIFAR10 and BIRD datasets using regular and saliency guided training. (C) Distribution of gradient means across examples. Maps produced by saliency guided training are more precise: most features have gradient values around zero with large gaps between mean and outliers. Here gradients around zero indicate uninformative features, while very large and very small gradients indicate informative features. Saliency guided training helps reduce noisy fluctuating gradients in between as shown in the box plots.

## Quality of Saliency Maps for Images

For an image classification problem, in many cases, most features are redundant and not needed by the model to make the prediction. Consider the background of an object in an image; although it covers most of the image, backgrounds are often not essential in the classification task. If the model is focusing on the object rather than the background, we would want the background gradient (i.e., most of the features) to be close to zero.

The examples shown in Figure 2 were correctly classified by both models. Gradients are scaled per sample to have values between -1 and 1. In Figure 2 (A) and Figure 2 (B), saliency maps produced by a model trained with saliency guided training were more precise than that trained traditionally. Most saliency maps produced by saliency guided training highlight the object itself rather than the background across different datasets. The distributions of gradient values per sample in Figure 2 (A) show that most features have small gradient values (near zero) with a large separation of high salient features away from zero for the saliency guided training. Similarly, in Figure 2 (C), we find that over the entire dataset, gradient values produced by the saliency guided training tend to be concentrated around zero with a large separation between the mean and outliers (highly salient features), indicating the model's ability to differentiate between informative and non-informative features.

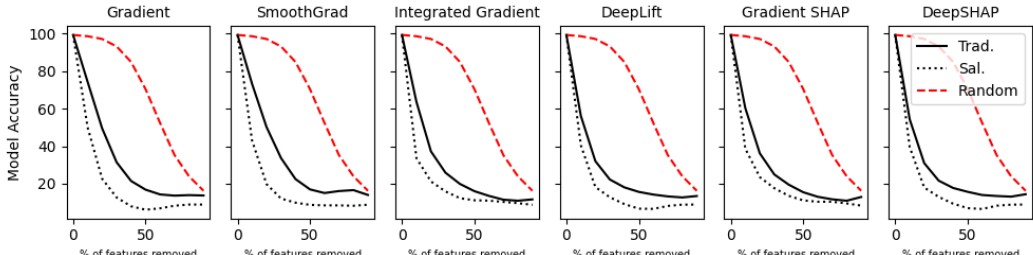

Figure 3: Model accuracy drop when removing features with high saliency using traditional and the saliency guided training for different gradient-based methods against a random baseline. A steeper drop indicates a better performance. We find that regardless of the saliency method used, the performance improves by the saliency guided training.

**Model Accuracy Drop**

We compare the saliency guided training and traditional training for different saliency methods with modification-based evaluation [45, 41, 23]: First, features are ranked according to the saliency values. Then, higher-ranked features are recursively eliminated (the original background in MNIST replaces the eliminated features). Finally, the degradation to the trained model accuracy is reported. This is done at different feature percentages. A steeper drop indicates that the removed features affected the model accuracy more. Figure 3 compares the model performance degradation on different gradient-based methods; the saliency guided training shows a steeper accuracy drop regardless of the saliency method used.

This experiment can only be performed on a dataset like MNIST since the uninformative feature distribution is known (black background), while this is not the case in other datasets that we have considered. Although such modification-based evaluation methods have been applied to other datasets, [45, 41, 23]; Hooker et al. [18] showed that removing features produces samples from a different data distribution violating the underlying IID assumption (i.e., the training and evaluation data come from identical distributions). When the feature replacement comes from a different distribution, it is unclear whether the degradation in the model performance is from the distribution shift or the removal of informative features. For that reason, we need to make sure that the model is trained on the mask used during testing to avoid this undesired effect.

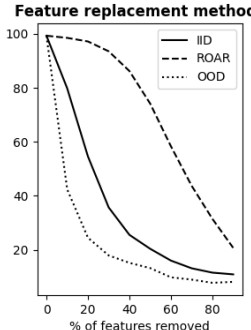

Figure 4: Accuracy drop in different modification-based evaluation masking approaches.

Hooker et al. [18] proposes ROAR where the model is retrained after the feature elimination. However, due to the data redundancy, the retrained model can rely on different features to achieve the same accuracy. Figure 4 shows the model accuracy drop on traditionally trained MNIST when removing the salient features. The IID line represents replacing features with the black MNIST background (known uninformative distribution), which acts as the ground truth in this particular dataset. The OOD line represents replacing the features with the mean image pixel value as done by [45, 41, 23]; and ROAR shows replacing features with the mean value and retraining the model as proposed by Hooker et al. [18]. Since neither OOD nor ROAR produce results similar to those produced by the IID feature replacement, we argue that modification-based evaluation methods may provide unreliable results unless the uninformative IID distribution is known. We leave further exploration of modification-based evaluation methods to future work.

## 5.2 Saliency Guided Training for Language

We compare the interpretability of recurrent models trained on language tasks using the ERASER [11] benchmark. ERASER was designed to capture how well an explanation provided by models aligns

with human rationales and how faithful these explanations are (i.e., the degree to which explanation influences the predictions). For our purpose, we only focus on the faithfulness of the explanations.

ERASER provides two metrics to measure interpretability. *Comprehensiveness* evaluates if all features needed to make a prediction are selected. To calculate an explanation comprehensiveness, a new input $\overline{X}_i$ is created such that $\overline{X}_i = X_i - R_i$ where $R_i$ is predicted rationales (i.e. the words selected by saliency method as informative). Let $f_\theta(X_i)_j$ be the prediction of model for class $j$. The model comprehensiveness is calculated as:

$$\text{Comprehensiveness} = f_\theta(X_i)_j - f_\theta(\overline{X}_i)_j \tag{5}$$

A high score here implies that the explanation removed was influential in the predictions. The second metric is *Sufficiency* that evaluates if the extracted explanations contain enough signal to make a prediction. The following equation gives the explanation sufficiency:

$$\text{Sufficiency} = f_\theta(X_i)_j - f_\theta(R_i)_j \tag{6}$$

A lower score implies that the explanations are adequate for a model prediction. The comprehensiveness and sufficiency were calculated at different percentages of features (similar to [11] percentages are $1\%, 5\%, 10\%, 20\%$ and $50\%$), and Area Over the Perturbation Curve (AOPC) is reported.

We focus on datasets that can be formulated as a classification problem: *Movie Reviews:* [62] positive/negative sentiment classification for movie reviews. *FEVER:* [55] a fact extraction and verification dataset where the goal is verifying claims from textual sources; each claim can either be supported or refuted. *e-SNLI:* [8] a natural language inference task where sentence pairs are labeled as entailment, contradiction, neutral and, supporting.

Word embeddings are generated from Glove [39]; then passed to a bidirectional LSTM [17] for classification. Table 1 compares the scores produced by different saliency methods for traditional and saliency guided training against random assignment baseline. We found that saliency guided training results in a significant improvement in both comprehensiveness and sufficiently for sentiment analysis task *Movie Reviews* dataset. While for fact extraction task *FEVER* dataset, and natural language inference task *e-SNLI* dataset saliency guided training improves comprehensiveness and there is no obvious improvement in sufficiency (this might be due to the adversarial effect of shrinking the sentence to a much smaller size since the number of words identified as "rationales" is smaller than the remaining words).

|  | Gradient | | Integrated Gradient | | SmoothGrad | | Random |
|  | Trad. | Sal. Guided | Trad. | Sal. Guided | Trad. | Sal. Guided | |
|---|---|---|---|---|---|---|---|
| **Movies** | | | | | | | |
| Comprehensiveness ↑ | 0.200 | *0.240* | 0.265 | ***0.306*** | 0.198 | *0.256* | 0.056 |
| Sufficiency ↓ | 0.042 | *0.013* | 0.054 | ***0.002*** | 0.034 | *0.008* | 0.294 |
| **FEVER** | | | | | | | |
| Comprehensiveness↑ | 0.007 | *0.008* | 0.008 | ***0.009*** | 0.007 | *0.008* | 0.001 |
| Sufficiency↓ | 0.012 | *0.011* | 0.005 | *0.004* | 0.006 | 0.006 | ***0.003*** |
| **e-SNLI** | | | | | | | |
| Comprehensiveness ↑ | 0.117 | ***0.126*** | 0.099 | *0.104* | 0.117 | *0.118* | 0.058 |
| Sufficiency↓ | 0.420 | *0.387* | 0.461 | *0.419* | 0.476 | *0.455* | ***0.366*** |

Table 1: Eraser benchmark scores: *Comprehensiveness* and *sufficiency* are in terms of AOPC. 'Random' is a baseline when words are assigned random scores.

## 5.3 Saliency Guided Training for Time Series

We evaluated saliency guided training for multivariate time series, both quality on multivariate time series MNIST and quantitatively through synthetic data.

**Saliency Maps Quality for Multivariate Time Series**

We compare the saliency maps produced on MNIST treated as a multivariate time series where one image axis is time. Figure 5 shows the saliency maps produced by different *(neural architecture, saliency method)* pairs when different training procedures were used. There is a visible improvement in saliency quality across different networks when saliency guided training is used.

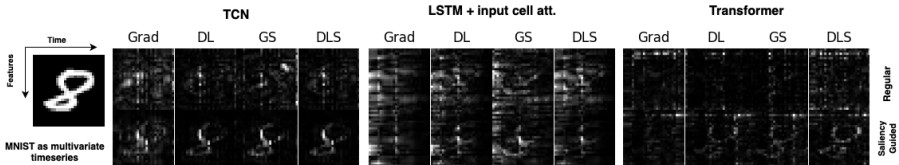

Figure 5: Saliency maps produced for *(neural architecture, saliency method)* pairs. Traditional training was used for networks in the 1st row, while saliency guided training was used for the 2nd row. Grad, DL, GS and DLS stand for Gradient, DeepLift, Gradient SHAP and DeepSHAP, respectively. There is an improvement in the quality of saliency maps when saliency guided training is used.

## Quantitative Analysis on Synthetic Data

We evaluated the saliency guided training on a multivariate time series benchmark proposed by Ismail et al. [21]. The benchmark consists of 10 synthetic datasets, each examining different design aspects in typical time series datasets. Informative features are highlighted by the addition of a constant $\mu$ to the positive class and subtraction of $\mu$ from the negative class. Following Ismail et al. [21], we compare 4 neural architectures: LSTM [17], LSTM with Input-Cell Attention [20], Temporal Convolutional Network (TCN) [28] and, Transformers [58]. Additional details about the dataset and architectures are provided in the supplementary material.

Quantitatively measuring the interpretability of a *(neural architecture, saliency method)* pair involves applying the saliency method, ranking features according to the saliency values, replacing high salient features with uninformative features from the original distribution at different percentages. Finally, the area under the precision curve (AUP) and the area under the recall curve (AUR) is calculated by the precision/recall values at different levels of degradation. Similar to Ismail et al. [21], we compare the AUP and AUR with a random baseline; since the baseline might be different for different models, we reported the difference between metrics values generated using the saliency method and the baseline. For example, the difference between gradient and random baseline *Diff*(AUP) when the model is trained traditionally is calculated as:

$$Diff(\text{AUP})_{Grad,Trad.} = AUP_{Grad,Trad.} - AUP_{Random,Trad.} \qquad (7)$$

Similarly difference when the model is trained using saliency guided training is:

$$Diff(\text{AUP})_{Grad,Sal.} = AUP_{Grad,Sal.} - AUP_{Random,Sal.} \qquad (8)$$

The mean metrics over all 10 datasets is shown in Table 2. Higher values indicate better performance; negative values indicate performance similar to random feature assignment. Overall, the best performance was achieved by *(TCN, Integrated gradients)* when using saliency guided training. Detailed results for each dataset are available in the supplementary material.

| Metric | Architecture | Gradient | | Integrated Gradient | | DeepLIFT | | Gradient SHAP | | DeepSHAP | | SmoothGrad | |
|---|---|---|---|---|---|---|---|---|---|---|---|---|---|
| | | Trad. | Sal. | Trad. | Sal. | Trad. | Sal. | Trad. | Sal. | Trad. | Sal. | Trad. | Sal. |
| *Diff*(AUP) | LSTM | -0.113 | -0.119 | -0.083 | -0.024 | -0.097 | -0.108 | -0.088 | -0.069 | -0.098 | -0.109 | -0.110 | -0.097 |
| | LSTM + Input. | 0.060 | *0.118* | 0.188 | *0.245* | 0.202 | *0.263* | 0.198 | *0.250* | 0.214 | ***0.272*** | 0.040 | *0.084* |
| | TCN | 0.106 | *0.168* | 0.233 | ***0.291*** | 0.248 | *0.270* | 0.235 | *0.288* | 0.263 | *0.280* | 0.088 | *0.155* |
| | Transformer | -0.054 | -0.062 | *0.061* | 0.044 | -0.040 | -0.032 | ***0.069*** | 0.023 | -0.014 | -0.055 | -0.018 | -0.046 |
| *Diff*(AUR) | LSTM | -0.017 | *0.019* | 0.062 | ***0.121*** | 0.047 | *0.089* | 0.060 | *0.102* | 0.031 | *0.075* | *0.007* | 0.004 |
| | LSTM + Input. | 0.075 | *0.136* | 0.185 | *0.198* | 0.187 | ***0.204*** | 0.182 | *0.196* | 0.183 | *0.201* | 0.043 | *0.111* |
| | TCN | 0.125 | *0.171* | 0.191 | ***0.210*** | 0.202 | *0.204* | 0.185 | *0.209* | *0.196* | 0.192 | 0.046 | *0.138* |
| | Transformer | 0.102 | *0.104* | ***0.182*** | 0.176 | 0.145 | *0.146* | 0.171 | 0.162 | *0.101* | 0.065 | *0.040* | 0.018 |

Table 2: The mean difference in weighted AUP and AUR for different *(neural architecture, saliency method)* pairs. Overall, the best preference was achieved by TCN when using Integrated gradients as a saliency method and saliency guided training procedure.

## Saliency Guided Training reduces vanishing saliency of recurrent neural networks

Ismail et al. [20] showed that saliency maps in RNNs vanish over time, biasing detection of salient features only to later time steps. This section investigates if using saliency guided training reduces the vanishing saliency issue in RNNs. Repeating experiments done by Ismail et al. [20], three

synthetic datasets were generated as shown Figure 6 (A). The specific features and the time intervals (boxes) on which they are considered important are varied between datasets to test the model's ability to capture importance at different time intervals. We trained an LSTM with traditional and saliency guided training procedures.

The area under precision curve (AUP) and the area under the recall curve (AUR) are calculated by the precision/recall values at different levels of degradation. Higher AUP and AUR suggest better performance. Results are shown in Figure 6 (B).

A traditionally trained LSTM shows clear bias in detecting features in the later time steps; AUP and AUR increase as informative features move to later time steps. When saliency guided training is used, LSTM was able to identify informative features regardless of their locations in time.

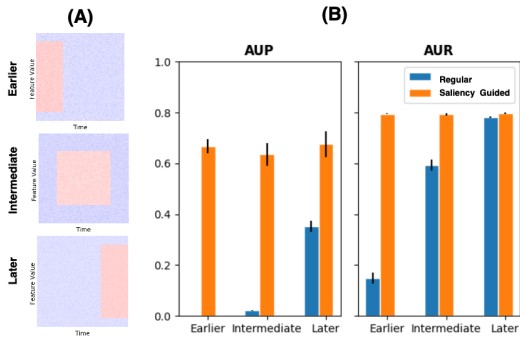

Figure 6: (A) Samples from 3 different simulated datasets, informative features are located at the earlier, intermediate, and later time steps. (B) AUP and AUR were produced by LSTM by traditional and saliency guided training procedures. Traditionally trained LSTM shows clear bias in detecting features in the later time steps. When saliency guided training is used, there is no time bias.

## 6  Summary and Conclusion

We propose *saliency guided training* as a new training procedure that improves the quality of explanations produced by existing gradient-based saliency methods. *saliency guided training* is optimized to reduce gradient values for irrelevant features. This is done by masking input features with low gradients and then minimizing the KL divergence between outputs from the original and masked inputs along with the main loss function. We demonstrated the effectiveness of the *saliency guided training* on images, language, and multivariate time series.

Our proposed training method encourages models to sharpen the gradient-based explanations they provide. It does this however without requiring explanations as input. It instead may be cast as a regularization procedure where regularization is provided by feature sparsity driven by a gradient-based feature attribution. This is an alternative approach to using ground truth explanations to force the model to be *right for the right reasons* [44]. We found that training model explanations in an unsupervised fashion also improves model faithfulness. This opens an interesting avenue for other unsupervised, perhaps regularization-based, methods to improve the interpretability of prediction models.

## 7  Acknowledgments

This project was supported in part by NSF CAREER AWARD 1942230, a grant from NIST 60NANB20D134, NSF award CDS&E:1854532, ONR grant 13370299 and AWS Machine Learning Research Award.

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
