# Supplementary Materials: Improving Deep Learning Interpretability by Saliency Guided Training

**Aya Abdelsalam Ismail, Héctor Corrada Bravo**\*, **Soheil Feizi** \*
{asalam,sfeizi}@cs.umd.edu, corradah@gene.com
Department of Computer Science, University of Maryland

## Experiments

**Computational resources:** All experiments were conducted on a single 12GB NVIDIA RTX2080Ti GPU. **Saliency Methods:** Captum [7] implementation was used for different saleincy methods.

### Saliency Guided Training for Images

#### Datasets and Classifiers

- **MNIST** [10]: a database of handwritten digits. The classifier consists of two CNN layers with kernel size 3 and stride of 1 followed by two fully connected layers, two dropout layers with $p = 0.25$ and $p = 0.5$, and the 10 output neurons.
- **CIFAR10** [8]: a low-resolution classification dataset with 10 different classes representing airplanes, cars, birds, cats, deer, dogs, frogs, horses, ships, and trucks. ResNet18 [3] was used as a classifier, ResNet18 is a deep CNN with "identity shortcut connection," i.e., skip connections, that skip one or more layers to solve the vanishing gradient problem faced by deep networks.
- **BIRD** [2]: A kaggle datasets of 260 bird species. Images were gathered from internet searches by species name. VGG16 [12] was used as a classifier, the last few dense layers and the output layer were modified to accommodate the number of classes in this dataset.

| Dataset | # Training | # Testing | # Classes | Features | Test Accuracy | | $\lambda$ | $k$ |
| --- | --- | --- | --- | --- | --- | --- | --- | --- |
| | | | | | Tradtional | Sal. Guided | | (as a % of feature) |
| MNIST | 60000 | 10000 | 10 | $1 \times 28 \times 28$ | 99.4 | 99.3 | 1 | 50% |
| CIFAR10 | 50000 | 10000 | 10 | $3 \times 32 \times 32$ | 92.0 | 91.5 | 1 | 50% |
| BIRD | 38518 | 1350 | 260 | $3 \times 224 \times 224$ | 96.6 | 96.9 | 1 | 50% |

Table 1: Datasets used for Image experiments. $k$ is the percentage of overall features masked during saliency guided training. For example, in MNIST number of features masked $\lceil 0.5 \times 28 \times 28 \rceil = 392$.

**Masking** For images, low salient features are replaced by a random variable within the color channel input range. For example, in an RGB image, if pixel $2 \times 3$ is to be masked $1 \times 2 \times 3$ would be replaced with a random variable within R channel range, similarly $2 \times 2 \times 3$ and $3 \times 2 \times 3$ would be replaced with a random variable within G and B channel range respectively.

### Saliency Map Quality for Images

The examples shown in Figure 1, Figure 2, and Figure 3 were correctly classified by both models. Gradients are scaled per sample to have values between -1 and 1. Overall, saliency maps produced

---

\*Authors contributed equally

35th Conference on Neural Information Processing Systems (NeurIPS 2021), Sydney, Australia.

by saliency guided training are less noisy than those produced by traditional training and tend to highlight the object itself rather than the background. The distributions of gradient values per sample show that most features have small values (near zero) with a higher separation of high saliency features away from zero for saliency guided training.

**Model Accuracy Drop**

We compare interpretable and traditional training for different saliency methods with modification-based evaluation. Each experiment is repeated five times. Figure 4 shows the mean and standard error for model degradation on different gradient-based methods.

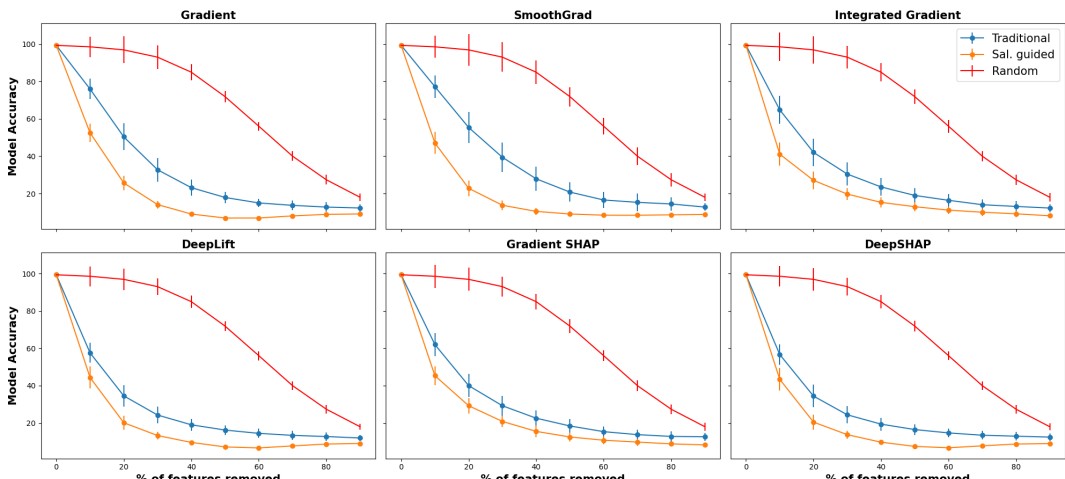

Figure 4: The mean and standard error for model accuracy drop when removing features with high saliency using traditional and saliency guided training for different gradient-based methods against a random baseline. A steeper drop indicates better performance. We find that regardless of the saliency method used, the performance improves by saliency guided training.

**Fine-tuning with Saliency guided Training**

We investigate the effect of training traditionally and fine-tuning with saliency guided training. This would be particularly useful for large datasets like imagenet. Table 2 shows the area under accuracy drop curve (AUC) on MNIST Figure 4 for gradient when training traditionally, training using saliency guided procedure and fine-tuning (smaller AUC indicates better performance). We find that fine-tuning improves the performance over traditionally trained networks.

| Training Procedures | AUC |
|---|---|
| Traditional | 3360.4 |
| Saliency Guided | 1817.6 |
| Fine-tuned | 2258.8 |

Table 2: Area under accuracy drop curve on MNIST for different training procedures

Note that, there is not much gain in training performance when training from scratch versus fine-tuning for small datasets like MNIST. However, for larger datasets like CIFAR10, we observed a clear decrease in the number of epochs when fine-tuning the network. The number of epochs for traditional training CIFAR10 is on average 118, saliency training is 124 while fine-tuning takes only 70 epochs.

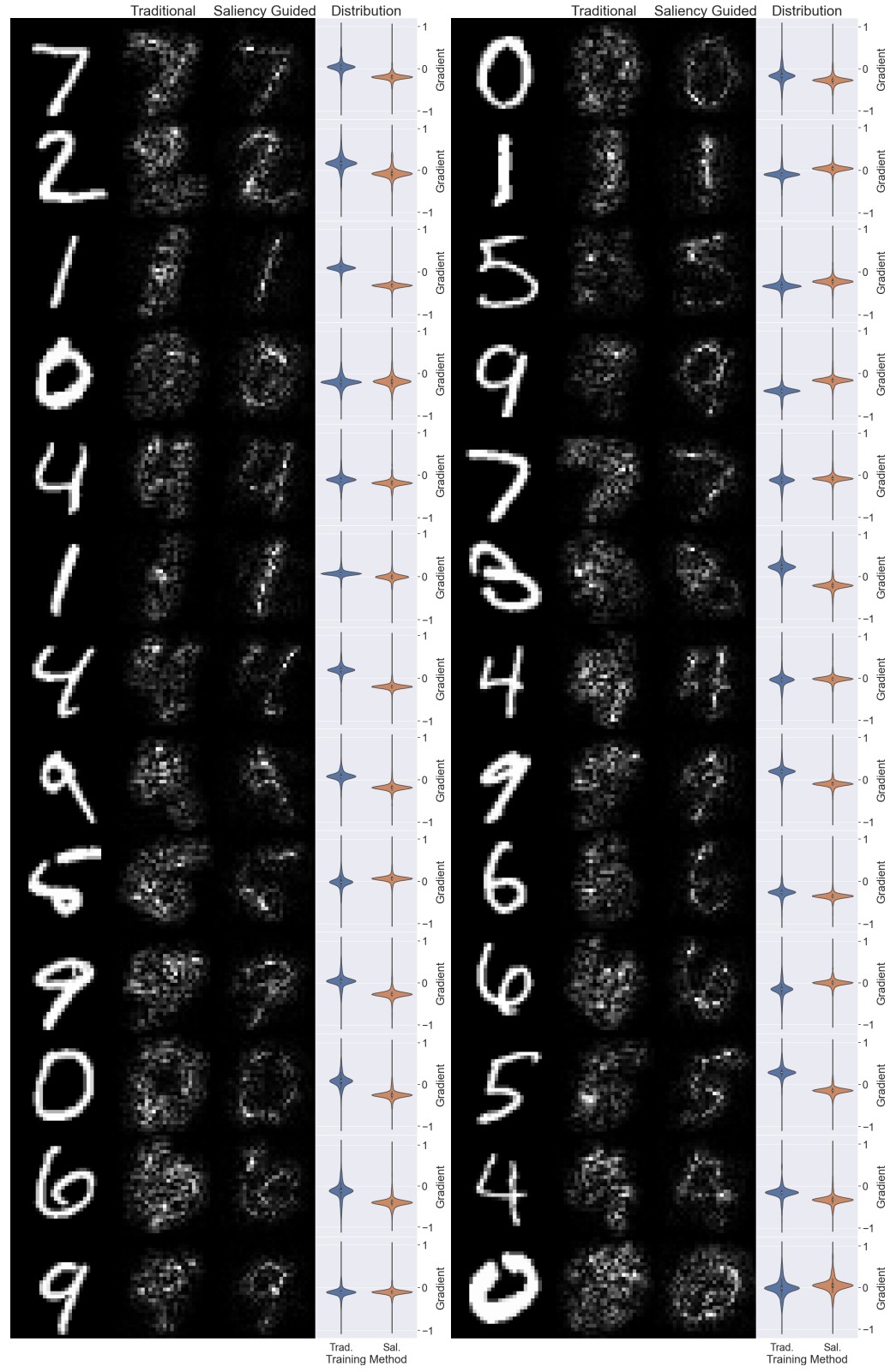

Figure 1: Saliency maps and saliency distribution for Traditional and Saliency Guided Training on MNIST

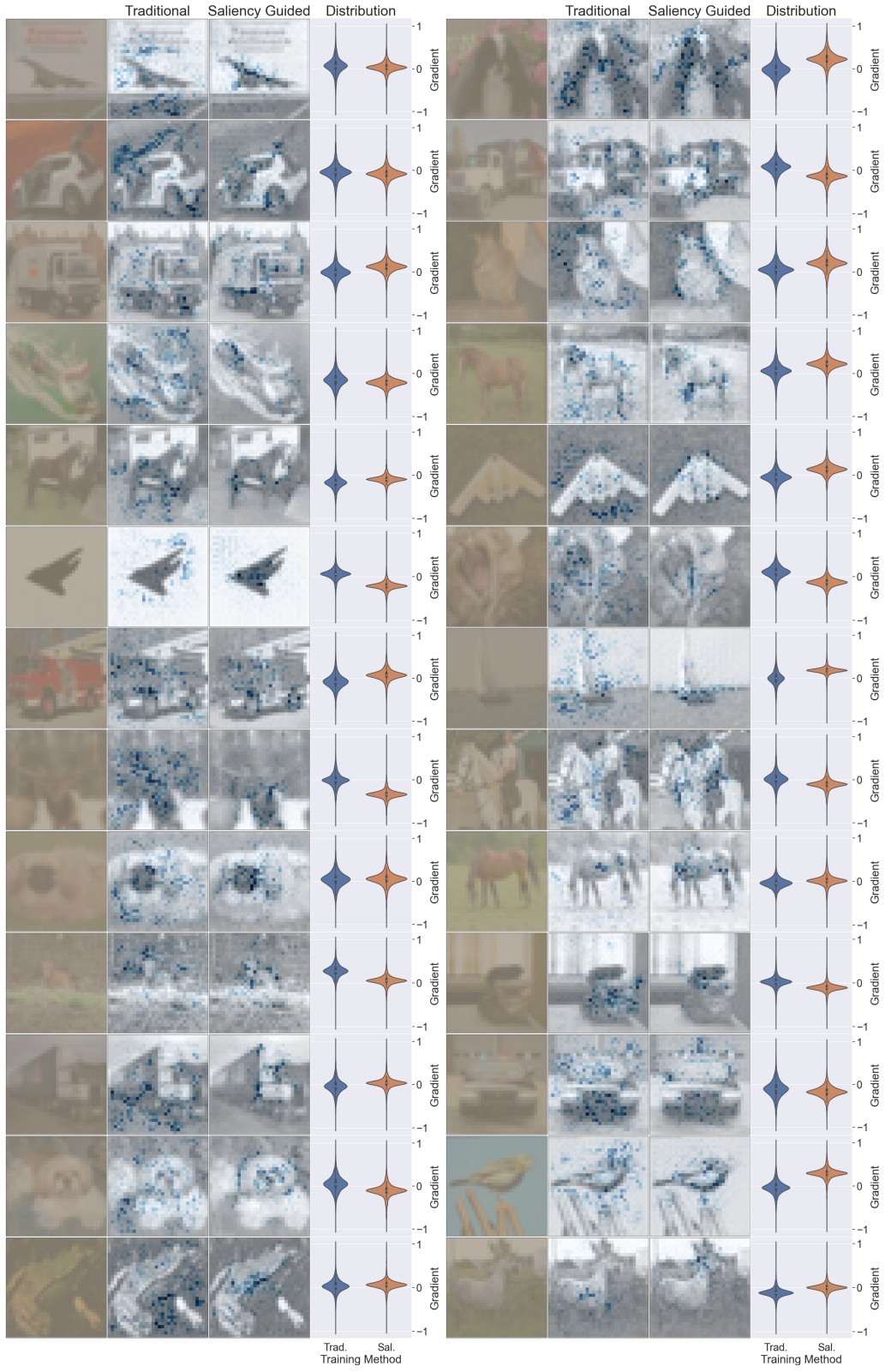

Figure 2: Saliency maps and saliency distribution for Traditional and Saliency Guided Training on CIFAR10

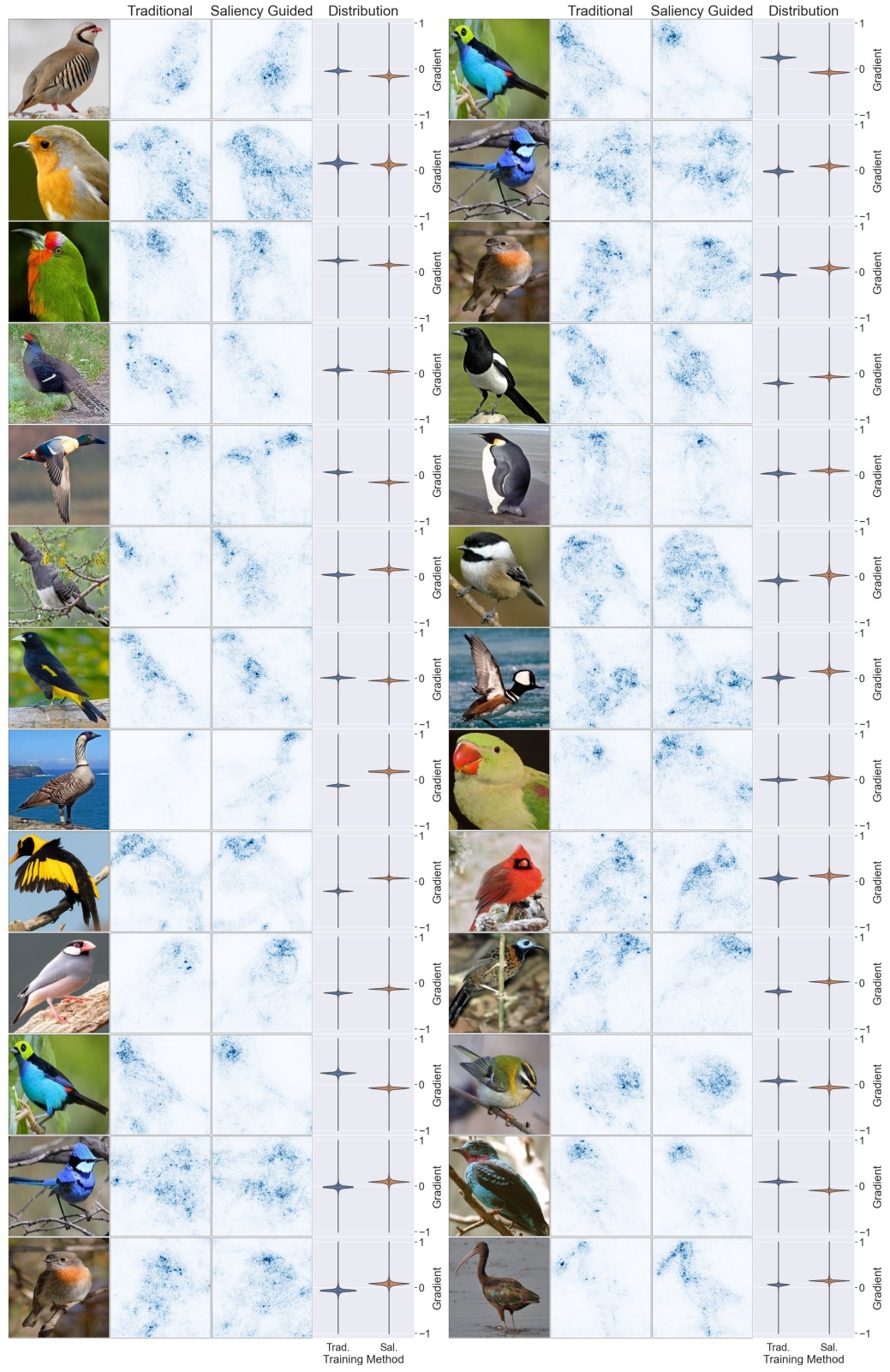

Figure 3: Saliency maps and saliency distribution for Traditional and Saliency Guided Training on BIRD

**Saliency Guided Training for Language**

We compare the interpretability of different models trained on language tasks using the ERASER [1] benchmark.

**Datasets**    For all datasets, words embbedding were generated from Glove [11] and a bidirectional LSTM [4] was used for classifications. Details about each dataset is available in Table 3

| Dataset | # Training | # Testing | # Classes | Tokens | Sentences | Test Accuracy | | $\lambda$ | $k$ |
| --- | --- | --- | --- | --- | --- | --- | --- | --- | --- |
| | | | | | | Tradtional | Sal. Guided | | (as a % of tokens) |
| Movie Review | 1600 | 200 | 2 | 774 | 36.8 | 0.8890 | 0.8980 | 1 | 60% |
| FEVER | 97957 | 6111 | 2 | 327 | 12.1 | 0.7234 | 0.7255 | 1 | 80% |
| e-SNLI | 911928 | 16429 | 3 | 16 | 1.7 | 0.9026 | 0.9068 | 1 | 70% |

Table 3: Overview of datasets in the ERASER benchmark. Number of labels, dataset size, and average numbers of sentences and tokens in each document. $k$ is the percentage of overall tokens within a particular document.

**Masking**    For language tasks, masking is a bit more tricky. We tried multiple masking function, including:

- **Removing** the masking function creates new input such that $\widetilde{X}$ contains only high salient word from the original input $X$.
- **Replace with token "[UNK]"** the masking function replaces the low salient word with the token "[UNK]" i.e., unknown.
- **Replace with token "[SEP]"** the masking function replaces the low salient word with the token "[SEP]" i.e., white space.
- **Replace with random word** the masking function replaces the low salient with a random word from vocabulary.
- **Replace with last high salient word** the masking function replaces the low salient word with the previous high salient word.

Over the three datasets, we found that the last masking function (replace with last high salient word) gave the best results. We believe that the masking function can also be dataset-dependent. This particular experiment aims to prove that saliency guided training improves interpretability on language tasks. We will consider finding the optimal masking function for different language tasks in our future work.

**Metrics**    ERASER provides two metrics to measure interpretability. *Comprehensiveness* evaluates if all features needed to make a prediction are selected. To calculate an explanation comprehensiveness, a new input $\overline{X}_i$ is created such that $\overline{X}_i = X_i - R_i$ where $R_i$ is predicted rationales. Let $f_\theta\left(X_i\right)_j$ be the prediction of model for class $j$. The model comprehensiveness is calculated as:

$$\text{Comprehensiveness} = f_\theta\left(X_i\right)_j - f_\theta\left(\overline{X}_i\right)_j$$

A high score here implies that the explanation removed was influential in the predictions. The second metric is *Sufficiency* that evaluates if the extracted explanations contain enough signal to make a prediction. The following equation gives the explanation sufficiency:

$$\text{Sufficiency} = f_\theta\left(X_i\right)_j - f_\theta\left(R_i\right)_j$$

A lower score implies that the explanations are adequate for a model prediction.

To evaluate the faithfulness of continuous importance scores assigned to tokens by models, the soft score over features provided by the model is converted into discrete rationales $R_i$ by taking the top-$k_d$ values, where $k_d$ is a threshold for dataset $d$. Denoting the tokens up to and including bin $k$, for instance, $i$ by $R_{ik}$, an aggregate *comprehensiveness* measure is defined as:

$$\frac{1}{|\mathcal{B}| + 1}\left(\sum_{k=0}^{|\mathcal{B}|} f_\theta\left(X_i\right)_j - f_\theta\left(\overline{X}_{ik}\right)_j\right)$$

*Sufficiency* is defined similarly. Here tokens are grouped into k = 5 bins by grouping them into the top $1\%, 5\%, 10\%, 20\%$ and $50\%$ of tokens, with respect to the corresponding importance score. This metrics is referred to as Area Over the Perturbation Curve (AOPC). For reference, we report these when random scores are assigned to tokens. Results are shown in the main paper Table 1.

## Saliency Guided Training for Time Series

We evaluated saliency guided training on a multivariate time series, both quality on multivariate time series MNIST and quantitatively through synthetic data.

## Saliency Maps Quality for Multivariate Time Series

We compare the saliency maps produced on MNIST treated as a multivariate time series with 28 time steps each having 28 features. Figure 5, Figure 6, and Figure 7 shows the saliency maps produced by different saliency methods for Temporal Convolutional Network (TCN), LSTM with Input-Cell Attention and, Transformers respectively. There is a visible improvement in saliency quality across different networks when saliency guided training is used. The most significant improvement was found in TCNs.

Figure 5: Saliency maps produced for *(TCN, saliency method)* pairs.

## Quantitative Analysis on Synthetic Data

We evaluated saliency guided training on a multivariate time series benchmark proposed by Ismail et al. [6]. The benchmark consists of 10 synthetic datasets, each examining different design aspects in typical time series datasets. Properties of each dataset is shown in Figure 8. Informative features are highlighted by the addition of a constant $\mu$ to the positive class and subtraction of $\mu$ from the negative class. For the following experiments $\mu = 1$. Details of each dataset is available in table 4.

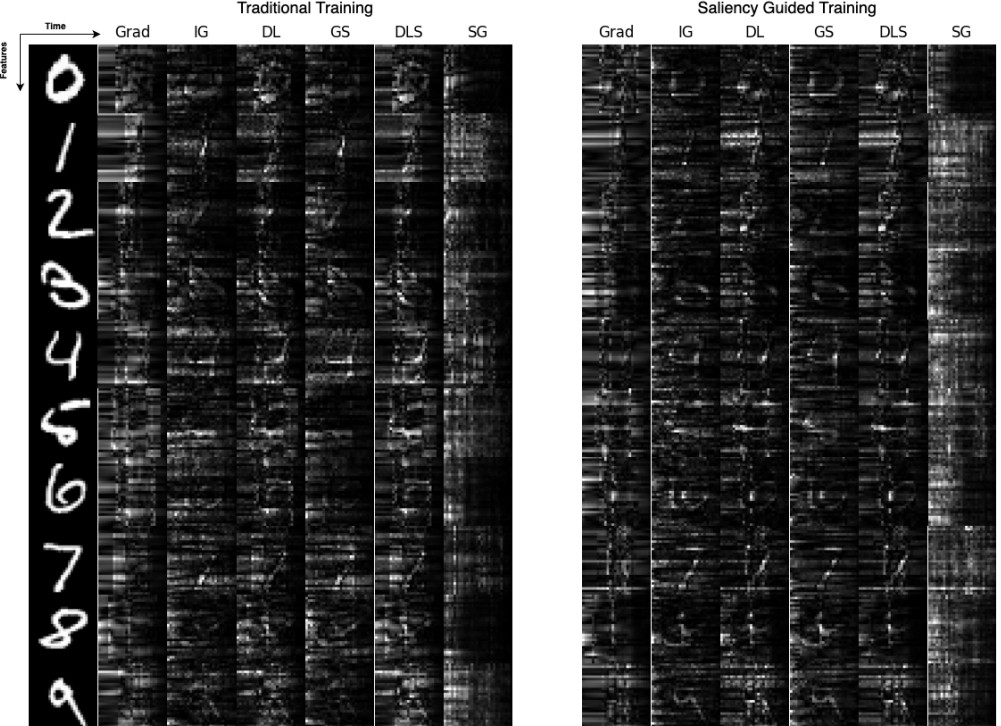

Figure 6: Saliency maps produced for *(LSTM with Input-Cell Attention, saliency method)* pairs.

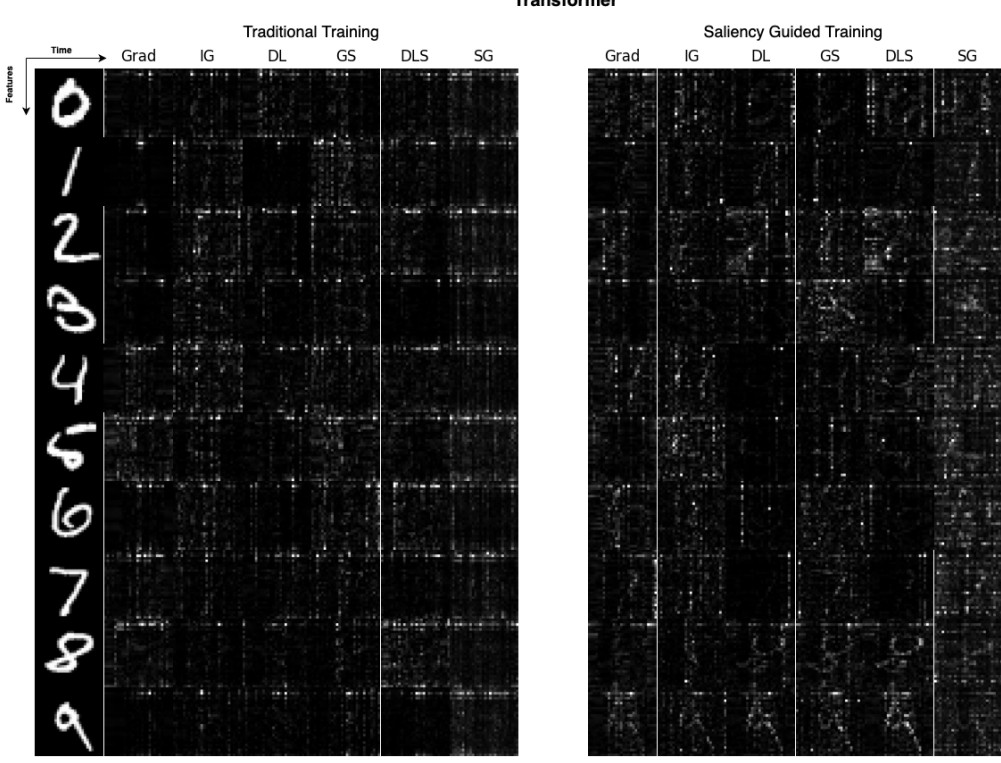

Figure 7: Saliency maps produced for *(Transformers, saliency method)* pairs.

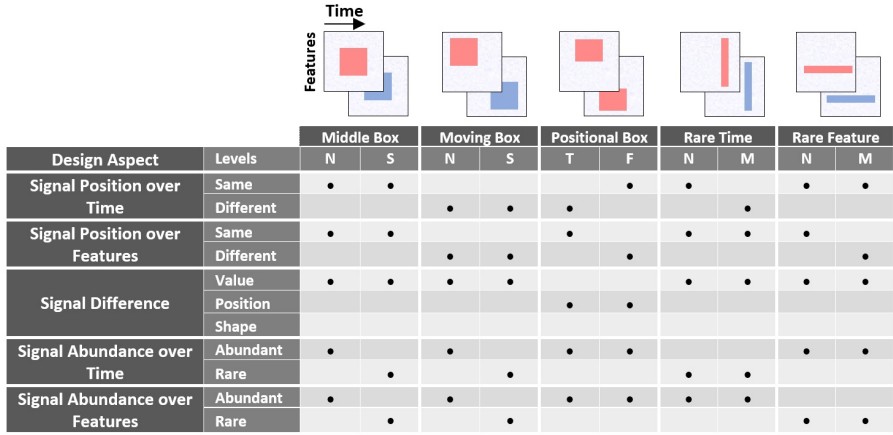

| Design Aspect | Levels | Middle Box | | Moving Box | | Positional Box | | Rare Time | | Rare Feature | |
|---|---|---|---|---|---|---|---|---|---|---|---|
| | | N | S | N | S | T | F | N | M | N | M |
| Signal Position over Time | Same | • | • | | | | • | • | | • | • |
| | Different | | | • | • | • | | | • | | • |
| Signal Position over Features | Same | • | • | | | • | | • | • | • | |
| | Different | | | • | • | | • | | | | • |
| Signal Difference | Value | • | • | • | • | | | • | • | • | • |
| | Position | | | | | • | • | | | | |
| | Shape | | | | | | | | | | |
| Signal Abundance over Time | Abundant | • | | • | | • | • | • | | • | • |
| | Rare | | • | | • | | | • | • | | |
| Signal Abundance over Features | Abundant | • | | • | | • | • | • | • | | |
| | Rare | | • | | • | | | | | • | • |

Figure 8: Figure from Ismail et al. [6]: Different evaluation datasets used for benchmarking saliency methods. Some datasets have multiple variations shown as sub-levels. N/S: normal and small shapes, T/F: temporal and feature positions, M: moving shape. All datasets are trained for binary classification. Examples are shown above each dataset, where dark red/blue shapes represent informative features.

| Dataset | # Training | # Testing | # Time Steps | # Feature | # Informative Time steps | # Informative Features |
|---|---|---|---|---|---|---|
| Middle | 1000 | 100 | 50 | 50 | 30 | 30 |
| Small Middle | 1000 | 100 | 50 | 50 | 15 | 15 |
| Moving Middle | 1000 | 100 | 50 | 50 | 30 | 30 |
| Moving Small Middle | 1000 | 100 | 50 | 50 | 15 | 15 |
| Rare Time | 1000 | 100 | 50 | 50 | 6 | 40 |
| Moving Rare Time | 1000 | 100 | 50 | 50 | 6 | 40 |
| Rare Features | 1000 | 100 | 50 | 50 | 40 | 6 |
| Moving Rare Features | 1000 | 100 | 50 | 50 | 40 | 6 |
| Postional Time | 1000 | 100 | 50 | 50 | 20 | 20 |
| Postional Feature | 1000 | 100 | 50 | 50 | 20 | 20 |

Table 4: Synthetic dataset details: Number of training samples, number of testing samples, number of time steps per sample, number of features per time step, number of time steps with informative features, and number of informative features in an informative time step.

Following Ismail et al. [6], we compare 4 neural architectures: LSTM [4], LSTM with Input-Cell Attention [5], Temporal Convolutional Network (TCN) [9] and, Transformers [13]. Each *(neural architecture, dataset)* pair was trained both traditionally and using saliency guided training. Test accuracy is reported in Table 5

| Datasets | LSTM | | LSTM+ Input-Cell | | TCN | | Transformer | |
|---|---|---|---|---|---|---|---|---|
| | Trad. | Sal. | Trad. | Sal. | Trad. | Sal. | Trad. | Sal. |
| Middle | 99 | 100 | 100 | 100 | 100 | 100 | 100 | 100 |
| Small Middle | 100 | 100 | 99 | 100 | 100 | 100 | 100 | 100 |
| Moving Middle | 100 | 100 | 100 | 100 | 100 | 100 | 99 | 100 |
| Moving Small Middle | 100 | 100 | 99 | 100 | 100 | 100 | 99 | 100 |
| Rare Time | 100 | 100 | 99 | 100 | 100 | 100 | 100 | 100 |
| Moving Rare Time | 100 | 100 | 100 | 100 | 100 | 100 | 99 | 100 |
| Rare Features | 100 | 100 | 100 | 100 | 100 | 100 | 100 | 100 |
| Moving Rare Features | 99 | 100 | 99 | 99 | 100 | 100 | 99 | 100 |
| Positional Time | 100 | 100 | 100 | 100 | 100 | 100 | 100 | 100 |
| Positional Feature | 100 | 100 | 99 | 100 | 99 | 100 | 100 | 100 |

Table 5: Test accuracy of different *(neural architecture, dataset)* pairs.

Quantitatively measuring the interpretability of a *(neural architecture, saliency method)* pair involves applying the saliency method, ranking features according to the saliency values, replacing high salient features with uninformative features from the original distribution at different percentages. Finally, we measure the model accuracy drop, weighted precision, and recall.

The area under precision curve (AUP) and the area under the recall curve (AUR) are calculated by the precision/recall values at different levels of degradation. Similar to Ismail et al. [6], we compare the AUP and AUR with a random baseline; since the baseline might be different for different models, we reported the difference between metrics values generated using the saliency method and the baseline. All experiments were ran 5 times the mean *Diff*(AUP), and *Diff*(AUR) is shown in Tables [6-9].

The results in Tables [6-9] show the follows: **LSTM**: Saliency guided training along with Integrated Gradient has the best precision and recall. **LSTM with Input Cell Attention**: Saliency guided training improves the performance of different saliency methods and datasets. DeepSHAP gives the best precision, while DeepSHAP gives the best recall. **TCN**: overall, saliency guided training improves the performance of different saliency methods and datasets. Integrated Gradient, Gradient SHAP, and DeepSHAP are best performing saliency methods. **Transformers**: have the worst interpretability. Using saliency guided training improved recall but not precision.

| Metric | Datasets | λ | k | Gradient Trad. | Gradient Sal. | Integrated Gradient Trad. | Integrated Gradient Sal. | DeepLIFT Trad. | DeepLIFT Sal. | Gradient SHAP Trad. | Gradient SHAP Sal. | DeepSHAP Trad. | DeepSHAP Sal. | SmoothGrad Trad. | SmoothGrad Sal. |
|---|---|---|---|---|---|---|---|---|---|---|---|---|---|---|---|
| *Diff*(AUP) | Middle | 1 | 30% | -0.280 | -0.280 | -0.261 | -0.036 | -0.267 | -0.270 | -0.263 | -0.124 | -0.267 | -0.271 | -0.283 | -0.269 |
| | Small Middle | 1 | 60% | -0.071 | -0.066 | -0.053 | 0.052 | -0.070 | -0.055 | -0.056 | -0.033 | -0.070 | -0.055 | -0.072 | -0.044 |
| | Moving Middle | 1 | 60% | -0.265 | -0.277 | -0.218 | -0.169 | -0.237 | -0.264 | -0.222 | -0.237 | -0.239 | -0.264 | -0.259 | -0.263 |
| | Moving Small Middle | 1 | 5% | -0.059 | -0.060 | -0.035 | 0.051 | -0.043 | -0.045 | -0.042 | -0.013 | -0.044 | -0.046 | -0.056 | -0.037 |
| | Rare Time | 1 | 30% | -0.076 | -0.076 | -0.075 | -0.065 | -0.076 | -0.075 | -0.075 | -0.071 | -0.076 | -0.076 | -0.076 | -0.068 |
| | Moving Rare Time | 1 | 50% | -0.067 | -0.058 | -0.042 | 0.016 | -0.053 | -0.042 | -0.045 | -0.010 | -0.054 | -0.043 | -0.061 | -0.032 |
| | Rare Feature | 1 | 30% | -0.063 | -0.075 | -0.039 | 0.006 | -0.047 | -0.073 | -0.038 | -0.027 | -0.048 | -0.073 | -0.069 | -0.059 |
| | Moving Rare Feature | 1 | 10% | -0.062 | -0.069 | -0.021 | 0.012 | -0.040 | -0.056 | -0.032 | -0.029 | -0.041 | -0.056 | -0.059 | -0.044 |
| | Postional Time | 1 | 30% | -0.116 | -0.119 | -0.040 | -0.006 | -0.107 | -0.112 | -0.058 | -0.046 | -0.108 | -0.113 | -0.111 | -0.102 |
| | Postional Feature | 1 | 2% | -0.064 | -0.104 | -0.042 | -0.104 | -0.028 | -0.089 | -0.043 | -0.105 | -0.031 | -0.091 | -0.055 | -0.053 |
| *Diff*(AUR) | Middle | 1 | 30% | 0.072 | 0.076 | 0.128 | 0.153 | 0.125 | 0.135 | 0.122 | 0.132 | 0.114 | 0.126 | 0.070 | 0.031 |
| | Small Middle | 1 | 60% | -0.043 | 0.037 | 0.048 | 0.157 | 0.029 | 0.116 | 0.038 | 0.129 | 0.007 | 0.102 | -0.032 | 0.009 |
| | Moving Middle | 1 | 60% | 0.060 | 0.073 | 0.119 | 0.124 | 0.110 | 0.124 | 0.099 | 0.117 | 0.099 | 0.115 | 0.061 | 0.042 |
| | Moving Small Middle | 1 | 5% | -0.032 | -0.004 | 0.046 | 0.135 | 0.043 | 0.073 | 0.042 | 0.093 | 0.025 | 0.060 | -0.023 | -0.025 |
| | Rare Time | 1 | 30% | -0.244 | -0.137 | -0.132 | 0.043 | -0.169 | -0.021 | -0.116 | 0.005 | -0.189 | -0.043 | -0.145 | -0.108 |
| | Moving Rare Time | 1 | 50% | -0.222 | -0.070 | -0.092 | 0.075 | -0.103 | 0.018 | -0.065 | 0.060 | -0.131 | 0.002 | -0.144 | -0.035 |
| | RareFeature | 1 | 30% | 0.182 | 0.197 | 0.219 | 0.218 | 0.217 | 0.223 | 0.216 | 0.216 | 0.211 | 0.219 | 0.191 | 0.166 |
| | Moving Rare Feature | 1 | 10% | 0.143 | 0.162 | 0.191 | 0.196 | 0.191 | 0.202 | 0.194 | 0.196 | 0.183 | 0.197 | 0.162 | 0.107 |
| | Postional Time | 1 | 30% | -0.032 | -0.073 | 0.072 | 0.119 | 0.029 | 0.021 | 0.046 | 0.082 | 0.012 | 0.001 | -0.019 | -0.064 |
| | Postional Feature | 1 | 2% | -0.053 | -0.070 | 0.016 | -0.005 | -0.002 | -0.005 | 0.004 | -0.009 | -0.018 | -0.025 | -0.056 | -0.083 |

Table 6: Difference in weighted AUP and AUR for *(LSTM, saliency method)* pairs. Overall, the best preference was achieved when using Integrated Gradients as a saliency method and saliency guided training as a training procedure.

| Metric | Datasets | λ | k | Gradient Trad. | Gradient Sal. | Integrated Gradient Trad. | Integrated Gradient Sal. | DeepLIFT Trad. | DeepLIFT Sal. | Gradient SHAP Trad. | Gradient SHAP Sal. | DeepSHAP Trad. | DeepSHAP Sal. | SmoothGrad Trad. | SmoothGrad Sal. |
|---|---|---|---|---|---|---|---|---|---|---|---|---|---|---|---|
| *Diff*(AUP) | Middle | 1 | 40% | 0.014 | 0.046 | 0.233 | 0.252 | 0.218 | 0.237 | 0.244 | 0.261 | 0.232 | 0.247 | -0.006 | 0.026 |
| | Small Middle | 1 | 60% | 0.049 | 0.150 | 0.161 | 0.273 | 0.169 | 0.305 | 0.170 | 0.273 | 0.180 | 0.312 | 0.038 | 0.091 |
| | Moving Middle | 1 | 80% | 0.044 | 0.082 | 0.262 | 0.251 | 0.260 | 0.256 | 0.276 | 0.256 | 0.277 | 0.261 | 0.010 | 0.044 |
| | Moving Small Middle | 1 | 5% | 0.044 | 0.055 | 0.181 | 0.201 | 0.179 | 0.196 | 0.187 | 0.200 | 0.190 | 0.204 | 0.022 | 0.029 |
| | Rare Time | 1 | 40% | 0.186 | 0.278 | 0.271 | 0.378 | 0.323 | 0.412 | 0.279 | 0.373 | 0.338 | 0.424 | 0.133 | 0.209 |
| | Moving Rare Time | 1 | 80% | 0.144 | 0.276 | 0.233 | 0.388 | 0.269 | 0.417 | 0.238 | 0.381 | 0.282 | 0.429 | 0.103 | 0.167 |
| | Rare Feature | 1 | 30% | 0.032 | 0.101 | 0.163 | 0.270 | 0.166 | 0.266 | 0.174 | 0.278 | 0.180 | 0.274 | 0.039 | 0.105 |
| | Moving Rare Feature | 1 | 5% | -0.002 | -0.004 | 0.120 | 0.124 | 0.116 | 0.116 | 0.124 | 0.127 | 0.126 | 0.126 | -0.003 | -0.004 |
| | Postional Time | 1 | 40% | 0.117 | 0.186 | 0.184 | 0.225 | 0.236 | 0.314 | 0.197 | 0.252 | 0.248 | 0.316 | 0.093 | 0.187 |
| | Postional Feature | 1 | 5% | -0.021 | 0.007 | 0.072 | 0.083 | 0.080 | 0.113 | 0.089 | 0.101 | 0.088 | 0.122 | -0.031 | -0.012 |
| *Diff*(AUR) | Middle | 1 | 40% | 0.028 | 0.084 | 0.163 | 0.176 | 0.160 | 0.180 | 0.162 | 0.173 | 0.157 | 0.177 | -0.001 | 0.044 |
| | Small Middle | 1 | 60% | 0.064 | 0.176 | 0.186 | 0.217 | 0.189 | 0.217 | 0.182 | 0.212 | 0.183 | 0.213 | 0.031 | 0.159 |
| | Moving Middle | 1 | 80% | 0.060 | 0.117 | 0.174 | 0.180 | 0.175 | 0.187 | 0.173 | 0.177 | 0.175 | 0.183 | 0.021 | 0.072 |
| | Moving Small Middle | 1 | 5% | 0.079 | 0.101 | 0.202 | 0.201 | 0.199 | 0.194 | 0.198 | 0.196 | 0.194 | 0.186 | 0.029 | 0.052 |
| | Rare Time | 1 | 40% | 0.139 | 0.203 | 0.214 | 0.225 | 0.214 | 0.233 | 0.211 | 0.223 | 0.211 | 0.233 | 0.103 | 0.191 |
| | Moving Rare Time | 1 | 80% | 0.118 | 0.213 | 0.198 | 0.226 | 0.200 | 0.233 | 0.193 | 0.224 | 0.194 | 0.232 | 0.070 | 0.197 |
| | RareFeature | 1 | 30% | 0.077 | 0.181 | 0.196 | 0.223 | 0.197 | 0.224 | 0.193 | 0.222 | 0.193 | 0.223 | 0.074 | 0.172 |
| | Moving Rare Feature | 1 | 5% | 0.059 | 0.039 | 0.188 | 0.189 | 0.191 | 0.186 | 0.182 | 0.183 | 0.186 | 0.180 | 0.038 | 0.028 |
| | Postional Time | 1 | 40% | 0.140 | 0.201 | 0.188 | 0.200 | 0.203 | 0.225 | 0.185 | 0.202 | 0.201 | 0.224 | 0.109 | 0.188 |
| | Postional Feature | 1 | 5% | -0.017 | 0.043 | 0.141 | 0.146 | 0.145 | 0.166 | 0.141 | 0.150 | 0.132 | 0.157 | -0.041 | 0.005 |

Table 7: The difference in weighted AUP and AUR for different *(LSTM with Input-Cell Attention, saliency method)* pairs. The use of saliency guided training improved the performance of most saliency methods. Overall, DeepSHAP and DeepLIFT produced the best precision and recall, respectively, when combined with saliency guided training.

| Metric | Datasets | λ | k | Gradient | | Integrated Gradient | | DeepLIFT | | Gradient SHAP | | DeepSHAP | | SmoothGrad | |
|---|---|---|---|---|---|---|---|---|---|---|---|---|---|---|---|
| | | | | Trad. | Sal. | Trad. | Sal. | Trad. | Sal. | Trad. | Sal. | Trad. | Sal. | Trad. | Sal. |
| *Diff*(AUP) | Middle | 1 | 50% | 0.127 | **0.217** | 0.283 | **0.393** | 0.350 | **0.398** | 0.290 | **0.384** | 0.365 | **0.416** | 0.090 | **0.194** |
| | Small Middle | 1 | 40% | 0.164 | **0.260** | 0.299 | **0.433** | 0.312 | **0.419** | 0.302 | **0.418** | 0.328 | **0.442** | 0.156 | **0.253** |
| | Moving Middle | 1 | 70% | 0.122 | **0.197** | 0.287 | **0.342** | 0.332 | **0.367** | 0.286 | **0.329** | 0.345 | **0.387** | 0.047 | **0.182** |
| | Moving Small Middle | 1 | 80% | **0.065** | 0.043 | **0.194** | 0.151 | 0.169 | **0.191** | **0.190** | 0.152 | 0.183 | **0.200** | **0.037** | 0.023 |
| | Rare Time | 1 | 50% | 0.184 | **0.290** | 0.314 | **0.363** | **0.324** | 0.309 | 0.314 | **0.360** | **0.352** | 0.319 | 0.177 | **0.226** |
| | Moving Rare Time | 1 | 50% | 0.142 | **0.182** | 0.260 | **0.333** | **0.257** | 0.243 | 0.258 | **0.330** | **0.275** | 0.251 | 0.122 | **0.179** |
| | Rare Feature | 1 | 30% | 0.058 | **0.244** | 0.246 | **0.451** | 0.252 | **0.422** | 0.249 | **0.453** | 0.286 | **0.450** | 0.085 | **0.259** |
| | Moving Rare Feature | 1 | 5% | -0.003 | **0.004** | 0.116 | **0.134** | 0.112 | **0.114** | 0.122 | **0.129** | 0.122 | **0.123** | **0.007** | 0.005 |
| | Postional Time | 1 | 70% | **0.115** | 0.072 | **0.180** | 0.114 | **0.233** | 0.069 | **0.187** | 0.117 | **0.237** | 0.035 | **0.106** | 0.066 |
| | Postional Feature | 1 | 10% | 0.082 | **0.176** | 0.151 | **0.199** | 0.136 | **0.162** | 0.155 | **0.203** | 0.137 | **0.175** | 0.058 | **0.159** |
| *Diff*(AUR) | Middle | 1 | 50% | 0.133 | **0.161** | 0.190 | **0.207** | 0.202 | **0.209** | 0.188 | **0.205** | 0.201 | **0.205** | 0.054 | **0.128** |
| | Small Middle | 1 | 40% | 0.086 | **0.230** | 0.194 | **0.240** | 0.202 | **0.240** | 0.189 | **0.239** | 0.196 | **0.241** | 0.039 | **0.230** |
| | Moving Middle | 1 | 70% | 0.134 | **0.144** | 0.191 | **0.195** | 0.201 | **0.208** | 0.186 | **0.194** | 0.201 | **0.203** | 0.036 | **0.121** |
| | Moving Small Middle | 1 | 80% | **0.118** | 0.117 | **0.204** | 0.199 | 0.193 | **0.195** | 0.196 | **0.196** | 0.186 | **0.190** | -0.001 | **0.065** |
| | Rare Time | 1 | 50% | 0.173 | **0.215** | 0.199 | **0.233** | **0.225** | 0.221 | 0.193 | **0.230** | **0.226** | 0.204 | 0.125 | **0.151** |
| | Moving Rare Time | 1 | 50% | 0.106 | **0.198** | 0.177 | **0.220** | **0.195** | 0.189 | 0.167 | **0.224** | **0.191** | 0.179 | -0.057 | **0.149** |
| | RareFeature | 1 | 30% | 0.152 | **0.222** | 0.222 | **0.239** | 0.223 | **0.239** | 0.219 | **0.239** | 0.224 | **0.239** | 0.130 | **0.217** |
| | Moving Rare Feature | 1 | 5% | 0.101 | **0.122** | 0.198 | **0.204** | **0.206** | 0.205 | 0.195 | **0.201** | 0.196 | **0.198** | 0.048 | **0.055** |
| | Postional Time | 1 | 70% | 0.126 | **0.128** | 0.156 | **0.172** | **0.194** | 0.160 | 0.147 | **0.165** | **0.181** | 0.110 | 0.039 | **0.102** |
| | Postional Feature | 1 | 10% | 0.126 | **0.174** | 0.177 | **0.196** | **0.180** | 0.175 | 0.172 | **0.194** | 0.164 | **0.154** | 0.049 | **0.160** |

Table 8: The difference in weighted AUP and AUR for different *(TCN, saliency method)* pairs. The use of saliency guided training improved the performance of most saliency methods. Overall, when combined with saliency guided training, Integrated Gradients and DeepSHAP produced the best precision. For recall, Integrated Gradients, DeepLift, Gradient SHAP, and DeepSHAP seem to perform similarly, again, the best performance was achieved when saliency guided training is used.

| Metric | Datasets | λ | k | Gradient | | Integrated Gradient | | DeepLIFT | | Gradient SHAP | | DeepSHAP | | SmoothGrad | |
|---|---|---|---|---|---|---|---|---|---|---|---|---|---|---|---|
| | | | | Trad. | Sal. | Trad. | Sal. | Trad. | Sal. | Trad. | Sal. | Trad. | Sal. | Trad. | Sal. |
| *Diff*(AUP) | Middle | 1 | 30% | -0.179 | -0.213 | **0.051** | -0.004 | -0.116 | -0.176 | **0.067** | -0.064 | -0.062 | -0.222 | -0.069 | -0.150 |
| | Small Middle | 1 | 60% | -0.034 | -0.057 | **0.054** | 0.042 | -0.018 | -0.034 | **0.066** | 0.024 | **0.009** | -0.060 | **0.006** | -0.022 |
| | Moving Middle | 1 | 90% | -0.188 | -0.146 | **0.062** | 0.018 | -0.142 | -0.067 | **0.065** | -0.011 | -0.130 | -0.143 | -0.091 | -0.157 |
| | Moving Small Middle | 1 | 70% | -0.002 | -0.008 | 0.031 | **0.039** | 0.017 | **0.029** | 0.026 | **0.037** | **0.036** | 0.016 | -0.021 | -0.035 |
| | Rare Time | 1 | 50% | **0.038** | -0.006 | **0.118** | 0.057 | -0.019 | **0.017** | **0.132** | 0.049 | **0.014** | -0.010 | -0.029 | -0.031 |
| | Moving Rare Time | 1 | 50% | **0.066** | 0.062 | **0.110** | 0.049 | -0.021 | **0.046** | **0.117** | 0.055 | -0.009 | **0.033** | -0.026 | -0.027 |
| | Rare Feature | 1 | 30% | -0.049 | -0.045 | 0.029 | **0.139** | -0.002 | -0.004 | 0.033 | **0.088** | 0.008 | -0.015 | 0.005 | **0.028** |
| | Moving Rare Feature | 1 | 10% | -0.034 | -0.031 | 0.041 | **0.055** | 0.008 | **0.014** | 0.038 | **0.049** | 0.008 | **0.022** | -0.003 | -0.013 |
| | Postional Time | 1 | 60% | -0.060 | -0.078 | **0.084** | 0.029 | -0.048 | -0.047 | **0.102** | -0.001 | **0.013** | -0.072 | **0.026** | -0.057 |
| | Postional Feature | 1 | 10% | -0.094 | -0.099 | **0.032** | 0.012 | -0.061 | -0.097 | **0.046** | 0.008 | -0.029 | -0.098 | **0.019** | 0.006 |
| *Diff*(AUR) | Middle | 1 | 30% | **0.087** | 0.053 | **0.167** | 0.146 | **0.155** | 0.112 | **0.157** | 0.111 | **0.119** | -0.051 | **0.040** | -0.025 |
| | Small Middle | 1 | 60% | **0.085** | 0.030 | 0.186 | **0.189** | **0.128** | 0.113 | **0.173** | 0.171 | **0.077** | -0.025 | **0.060** | 0.036 |
| | Moving Middle | 1 | 90% | **0.071** | 0.134 | 0.164 | **0.185** | 0.136 | **0.181** | 0.150 | **0.183** | 0.057 | **0.130** | 0.019 | **0.040** |
| | Moving Small Middle | 1 | 70% | 0.118 | **0.137** | 0.171 | **0.177** | 0.157 | **0.171** | 0.160 | **0.171** | 0.098 | **0.117** | -0.004 | -0.019 |
| | Rare Time | 1 | 50% | 0.152 | **0.139** | **0.206** | 0.172 | 0.116 | **0.135** | **0.199** | 0.157 | 0.077 | **0.088** | -0.027 | -0.073 |
| | Moving Rare Time | 1 | 50% | 0.184 | **0.185** | **0.198** | 0.170 | 0.124 | **0.175** | **0.186** | 0.168 | 0.059 | **0.127** | -0.033 | -0.013 |
| | RareFeature | 1 | 30% | 0.115 | **0.152** | 0.184 | **0.217** | 0.187 | **0.188** | 0.173 | **0.203** | 0.144 | **0.149** | 0.087 | **0.135** |
| | Moving Rare Feature | 1 | 10% | 0.101 | **0.122** | 0.179 | **0.183** | 0.174 | **0.180** | 0.165 | **0.176** | 0.125 | **0.149** | 0.060 | 0.034 |
| | Postional Time | 1 | 60% | **0.091** | 0.071 | **0.193** | 0.172 | **0.154** | 0.150 | **0.184** | 0.151 | **0.145** | 0.059 | **0.103** | -0.004 |
| | Postional Feature | 1 | 10% | 0.017 | 0.013 | 0.170 | **0.149** | **0.123** | 0.057 | **0.160** | 0.131 | **0.105** | -0.072 | **0.094** | 0.073 |

Table 9: The difference in weighted AUP and AUR for different *(Transformers, saliency method)* pairs. In this benchmark, Transformers seem to have the worst interpretability. Using saliency guided training improved recall but not precision. Overall best precision was achieved when combining traditional training with Gradient SHAP. While best recall was achieved when using saliency guided training and Integrated Gradients.