# OpenReview forum: "Improving Deep Learning Interpretability by Saliency Guided Training"
_NeurIPS.cc/2021/Conference — NeurIPS 2021 Poster_

### Official Review · Reviewer_ooFj · 2021-07-09

**Rating:** 8
**Confidence:** 4

**Summary:**

This paper addresses the noise in saliency map techniques, which seek to explain what part of the input is responsible for a model's output decision, by introducing a novel training paradigm in which noise is added to the least important input features and the model is encouraged to produce similar outputs for original and noise-augmented inputs. This paper shows extensive experimental results in a variety of ML domains, such as image, language, and time series data. Additionally, the paper adds interesting commentary on input feature replacement paradigms such as ROAR and OOD as well as analysis that shows how their interpretable training paradigm reduces the vanishing saliency problem in RNNs.

**Limitations And Societal Impact:**

The authors briefly mention some basic limitations of their method (computational cost, hyperparameter tuning, requiring training); however, they can improve their limitations section by fleshing out in more detail the additional computational cost. Furthermore, they did not discuss potential negative impacts of their work. Some directions for discussing potential negative impacts include misusing saliency methods, lack of consensus on how to evaluate saliency methods, etc.

**Main Review:**

This paper is well motivated -- a number of saliency methods are affected by noisy gradients, often mitigating this effect with solutions that aren't well motivated (e.g., changing backprop rules). The paper is extensive, thoughtful, and thorough in its experimentation and analysis, covering a wide range of ML domains and showing convincingly that their paradigm works across different settings, for instance, by reducing the noise in saliency maps, widening the separation between uninformative and informative input features as highlighted by saliency methods, improving performance of saliency map methods when evaluated on the model accuracy drop benchmark. In particular, its analysis on feature replacement paradigms (section 5.1, under "model accuracy drop") a useful insight into how to consider methods like ROAR and OOD. Additionally, its experiment showing that its interpretable training paradigm mitigates the vanishing saliency of recurrent neural networks is an exciting result. Both the insights presented in this paper as well as the novel training paradigm introduced can be used and built upon in future works.

The rest of this review contains large and small suggestions for the authors to consider.

Large:
* Could this paradigm be adapted and used as a fine-tuning procedure? Would we see the same benefits (for less training time)?
* How does this training paradigm impact performance (e.g., accuracy)? Accuracy numbers provided for language but not other domains
* Figure 2A -- distribution plots are hard to interpret -- use the box plot versions for MNIST (as shown in supp mat)
* Table 1 -- include additional results when using other saliency methods as well as LSTM Interpretable + other saliency methods

Small:
* Additional related works from weak localization literature that use input-level perturbations during training: Wei et al., CVPR 2017 (Object Region Mining with Adversarial Erasure), Singh & Lee, ICCV 2017 (Hide-and-Seek), DeVries 2017 (Cutout).
* Line 119: Expand a bit more on what additional computational cost is needed (does it take the same # of epochs to converge; what's the added memory cost?)
* Explain more clearly why we should expect feature replacement methods to match IID replacement
* Line 114: Be more explicit that only input features are being masked (e.g., not intermediate features), be more clear in main paper about what masked features are being replaced by (e.g., copy parts from supp mat)
* Line 132: Spell out which VGG network is used (e.g., VGG-16?)
* Line 186: \tilde{X} is already used on line 103 - use different notation for clarity
* Line 186: Explain what R_i (predicted rationales) are in this context
* Figure 5 -- in caption, explain what the abbreviations for saliency methods mean (Grad, DL, GS, DLS - last few are a bit unclear)


**Time Spent Reviewing:**

2

---

> ### Author Response · Authors · 2021-08-10
> **Response to Reviewer ooFj**
>
> Thank you for your thoughtful and detailed review.
>
> ***Re: “Could this paradigm be adapted and used as a fine-tuning procedure?”*** Yes and this would be particularly useful for large datasets like imagenet. Below are some quantitative results showing the area under accuracy drop curve on MNIST (figure  3 in main paper) for gradient when training traditionally, training using interpretable procedure and fine-tuning (smaller AUC indicates better performance).
> We find that fine-tuning improves the performance over traditionally trained networks. There is not much gain in training performance when training from scratch versus fine-tuning for small datasets like MNIST; however, for larger datasets like CIFAR10 we observed a clear decrease in the number of epochs when fine-tuning the network. The number of epochs for traditional training CIFAR10 is on average 118, interpretable training is 124 while fine-tuning takes only 70 epochs.
>
> |               | AUC     |
> |---------------|---------|
> | Traditional   | 3360.4  |
> | Interpretable | 1817.6  |
> | Fine-tuned    | 2258.75 |
>
>
> ***Re: “How does this training paradigm impact performance”*** There is minimal impact on the model accuracy. We will add this to the finial draft
> Below are the results from vision datasets:
>
> |         | Traditional Acc. | Interpretable Acc. |
> |---------|--------------|--------------------|
> | MNIST   |  99.4            |     99.3               |
> | CIFAR10 |   92.0           |    91.5                |
> | BIRD    |     96.6          |    96.9                |
>
> Below are the results from time series datasets:
>
> |                      | LSTM Trad. | LSTM Int. | LSTM+ Input-Cell Trad. | LSTM+ Input-Cell Int. | TCN Trad. | TCN Int. | Transformer Trad. | Transformer Int. |
> |----------------------|------------|-----------|------------------------|-----------------------|-----------|----------|-------------------|------------------|
> | Middle               | 99         | 100       | 100                    | 100                   | 100       | 100      | 100               | 100              |
> | Small Middle         | 100        | 100       | 99                     | 100                   | 100       | 100      | 100               | 100              |
> | Moving Middle        | 100        | 100       | 100                    | 100                   | 100       | 100      | 99                | 100              |
> | Moving Small Middle  | 100        | 100       | 99                     | 100                   | 100       | 100      | 99                | 100              |
> | Rare Time            | 100        | 100       | 99                     | 100                   | 100       | 100      | 100               | 100              |
> | Moving Rare Time     | 100        | 100       | 100                    | 100                   | 100       | 100      | 99                | 100              |
> | Rare Features        | 100        | 100       | 100                    | 100                   | 100       | 100      | 100               | 100              |
> | Moving Rare Features | 99         | 100       | 99                     | 99                    | 100       | 100      | 99                | 100              |
> | Positional Time       | 100        | 100       | 100                    | 100                   | 100       | 100      | 100               | 100              |
> | Positional Feature    | 100        | 100       | 99                     | 100                   | 99        | 100      | 100               | 100              |
>
>
>
>
> ***Re: “Figure 2A -- distribution plots are hard to interpret”*** Thank you for pointing this out. We will improve this visualization and use the box plot in the supplementary.
>
>
>
> ***Re: “Table 1 -- include additional results when using other saliency methods”***  As per your request, we will include other saliency methods. Results for integrated gradients and SmoothGrad are shown below. Overall, the use of interpretable training improves the comprehensiveness for different saliency methods. We will add them along with others in the final draft.
>
> |                                     | Comprehensiveness | Sufficiency |
> |-------------------------------------|-------------------|-------------|
> | Movies                              |                   |             |
> | Glove+LSTM + IG                     | 0.265             | 0.054       |
> | Glove+LSTM Interpretable + IG       | **0.306**             | **0.002**       |
> | Glove+LSTM + SmoothGrad               |   0.198                 |   0.034          |
> | Glove+LSTM Interpretable + SmoothGrad |   **0.256**                |  **0.008**           |
> | Fever                               |                   |             |
> | Glove+LSTM + IG                     |     0.008             |    0.005         |
> | Glove+LSTM Interpretable + IG       |  **0.009**            |  **0.004**           |
> | Glove+LSTM + SmoothGrad               |     0.007              |    0.006         |
> | Glove+LSTM Interpretable + SmoothGrad |    **0.008**               |   0.006          |
> | e-SNLI                              |                   |             |
> | Glove+LSTM + IG                     |   0.099                |   0.461          |
> | Glove+LSTM Interpretable + IG       |  **0.104**                  |  **0.419**            |
> | Glove+LSTM + SmoothGrad               |     0.117              |   0.476          |
> | Glove+LSTM Interpretable + SmoothGrad |  **0.118**                |    **0.455**
>
>
> ***Re: “Additional related works from weak localization literature”*** Thank you for pointing this out. We will add the mentioned papers along with others in the related work.
>
>
> ***Re: “Line 119: Expand a bit more on what additional computational cost”*** The memory needed is doubled since now in addition to storing the batch we are storing the masked batch as well. Similar to adversarial training, this training process is slow and takes a larger number of epochs to converge. For example, CIFAR10 when traditionally trained takes on average 118 epochs to converge each epoch is roughly 24 seconds. While using interpretable training takes 124 epochs to converge each epoch taking 75 seconds (all experiments on the same GPU).
> We will mention this explicitly in the updated paper.
>
>
> ***Re: “Explain more clearly why we should expect feature replacement methods to match IID replacement”***
> To get accurate results during testing, one key assumption in machine learning is that the training and evaluation data come from the same distribution. When feature replacement comes from a different distribution, it is unclear whether the degradation in model performance comes from either the distribution shift or because the features that were removed were informative. For that reason, we need to make sure that the model is trained on the mask used during testing to avoid this adversarial effect.
>
>
>
> ***Re: “Line 114: Be more explicit that only input features are being masked”*** We will clarify this in the main text and add details from the supplementary as advised.
>
>
>
> ***Re: “Line 132: Spell out which VGG network is used ”*** You are correct that we are using VGG16. We will explicitly mention this.
>
>
>
> ***Re: “Line 186: \tilde{X} is already used”*** Thank you for catching this. We will update this in the final draft.
>
>
> ***Re: “Line 186: Explain what R_i (predicted rationales) are in this context”*** Predicted rationales are the words selected by the saliency method as informative. We will clarify this in the final draft.
>
>
> ***Re: “Figure 5 -- in caption, explain what the abbreviations for saliency methods mean  ”*** Grad, DL, GS, DLS stand for gradient, DeepLift, Gradient SHAP and DeepLift with Shapely values, respectively. We will explicitly mention them in figure 5.

---

> > ### Comment · Reviewer_ooFj · 2021-08-23
> > **Thank you**
> >
> > Thank you for your thorough rebuttal, especially in providing details of rebuttal experiments and clarifying unclear writing. Based on the rebuttal and other reviews, I plan to keep my original rating of 8 (top 50% accept).

---

### Official Review · Reviewer_KCTi · 2021-07-12

**Rating:** 6
**Confidence:** 4

**Summary:**

This paper proposes a method (interpretable training) to improve the interpretability of saliency methods. The interpretable training method iteratively masks the features with small vanilla gradients and maximizes the similarity of the model outputs for both masked and unmasked inputs. They demonstrate this training procedure can improve interpretability in various data sets from CV, NLP, and time series across neural architectures, including RNNs, CNNs, and Transformers. They also find this method can reduce the vanishing saliency issue of RNNs. Lastly, they demonstrate improvements of the explanations produced after training by several gradient-based saliency methods.

**Main Review:**

Originality: Based on an intuitive idea that if gradient-based explanation methods faithfully interpret the model's predictions, irrelevant features should have gradient values close to zero, the authors provide a method (interpretable training) to improve interpretability. The proposed method combines the masking approach (line 103) and the KL divergence regularization (line 97) to solve the optimization problem in different tasks. The authors formulated this interpretable training with a concise equation of loss function (i.e., the prediction loss and the KL loss, line 107). The algorithm for interpretable training (Algorithm 1) is simple but novel for computing the masked input in each minibatch. Nevertheless, the reviewer thinks it is an effective approach to improve explainability.

Presentation Quality: The authors give extensive empirical analysis, especially for CV tasks (i.e., qualitative analysis in Figures 2A and 2B, quantitative analysis in Figure 2C and Figure 3), to demonstrate the better performance for interpretable training compared with other regular training methods. However, the following substantial issues should be addressed: (1) In Table 1, the performance for interpretable training for NLP tasks does not demonstrate consistent performance; the reason why a lower sufficiency for both FEVER and e-SNLI is not explained.  (2) In Table 1, for NLP tasks, only the vanilla gradient-based method is used; how about the more popular ones like DeepLift or IG? (3) In lines 114-116, the authors give how to implement masking functions in image, language, but how about for the time series?  Why can't these operations replace the low salient word with the previous high salient word (line 115) guarantee the argument of assigning low gradient values to irrelevant features in model predictions (in line 112), any theoretical explanation?  (4) Since the parameter k is very important for masking operation, in lines 116 -117, the authors point out how to select k for images (MNIST dataset); however, the selection of k for language and time series is not clear. (5) Other minor problems, e.g., the sequence number should be added for other formulas other than the KL-divergence (Eq (1)),  line number for Algorithm 1.

-----New comments after reading authors' response

I appreciate the response given by the authors, and most of my concerns have been successfully addressed by the responses. I think this work is  novel compared to other existing works on "interpretable training" in that this method can be used in more broad areas, NLP, CV, and time series prediction tasks, which is achieved via masking gradients without using attention maps. As such, I raise my rating to 6 to reflect my new assessment of the updated manuscript.




**Time Spent Reviewing:**

6

---

> ### Author Response · Authors · 2021-08-10
> **Response to Reviewer KCTi**
>
> Thank you for your detailed review.
>
>
> ***Re: “(1) In Table 1, the performance for interpretable training for NLP tasks does not demonstrate consistent performance”*** It was surprising to find that generally, the sufficiency of different saliency methods is similar (or even in some cases worse) than random assignment, this is the case regardless of the training method. One possible explanation is due to the adversarial effect of shrinking the sentence to a much smaller size since the number of words identified as “rationales” is smaller than the remaining words. However, we decided to report both metrics as done in the original paper.
>
> ***Re: “ (2) In Table 1, for NLP tasks only the vanilla gradient-based method is used, how about the more popular ones*** As per your request, we added integrated gradients and SmoothGrad; results are shown below.
> Overall, the use of interpretable training improves the comprehensiveness for different saliency methods.
> We did not include DeepLift in the table below due to time constraints as it is non trivial to incorporate DeepLift in the Eraser benchmark.
> We will add these methods along with others in the final draft.
>
> |                                     | Comprehensiveness | Sufficiency |
> |-------------------------------------|-------------------|-------------|
> | Movies                              |                   |             |
> | Glove+LSTM + IG                     | 0.265             | 0.054       |
> | Glove+LSTM Interpretable + IG       | **0.306**             | **0.002**       |
> | Glove+LSTM + SmoothGrad               |   0.198                 |   0.034          |
> | Glove+LSTM Interpretable + SmoothGrad |   **0.256**                |  **0.008**           |
> | Fever                               |                   |             |
> | Glove+LSTM + IG                     |     0.008             |    0.005         |
> | Glove+LSTM Interpretable + IG       |  **0.009**            |  **0.004**           |
> | Glove+LSTM + SmoothGrad               |     0.007              |    0.006         |
> | Glove+LSTM Interpretable + SmoothGrad |    **0.008**               |   0.006          |
> | e-SNLI                              |                   |             |
> | Glove+LSTM + IG                     |   0.099                |   0.461          |
> | Glove+LSTM Interpretable + IG       |  **0.104**                  |  **0.419**            |
> | Glove+LSTM + SmoothGrad               |     0.117              |   0.476          |
> | Glove+LSTM Interpretable + SmoothGrad |  **0.118**                |    **0.455**
>
> ***Re: “(3) In lines 114-116 the authors give how to implement masking functions”*** For time series, similar to images, masking is done by replacing each feature with a random variable within the feature range. For NLP, we replaced low salient features with previous high salient ones because we want to emphasize the salient features and remove the non-salient ones while maintaining the sentence length. We will add these details to the updated draft.
>
> ***Re: “(4) Since the parameter k is very important for masking operation*** K is generally dataset-dependent. K was optimized on a validation dataset. The value of K depends on the amount of irrelevant information in a training sample. For example, in the NLP dataset Movies review the average number of tokens per review is 774.  Below is an example from the movie review dataset.
>
> > Kidman is really the only thing that 's worth watching in Birthday Girl , a film by the stage-trained Jez Butterworth -LRB- Mojo -RRB- that serves as yet another example of the sad decline of British comedies in the post-Full Monty world .
>
>
> While in the e-SNLI dataset the average number of tokens is 16. A data sample is shown below:
>
> > Premise: An adult dressed in black holds a stick.
> Hypothesis: An adult is walking away, empty-handed.
>
>
> Movie reviews tend to have more filler words, so K for movie reviews (on average 465) is larger than that of e-SNLI ( on average 11).
>
> ***Re: “ (5) Other minor problems”*** Thank you for pointing this out. We will adjust this in the final draft.

---

### Official Review · Reviewer_jiTu · 2021-07-16

**Rating:** 6
**Confidence:** 4

**Summary:**

The authors propose an interpretable training procedure, in which the input features with the bottom k input gradients are masked. The loss term is comprised of the original task loss, plus a KL divergence between the model outputs on the original and masked inputs. The authors perform experiments on 3 image datasets, 3 language datasets, and a suite of synthetic time series datasets. They use model accuracy drop as a metric to evaluate saliency map quality.

**Limitations And Societal Impact:**

The authors address the limitations of their method around Line 119, about the additional computation needed for training and about the extra hyperparameters their method introduces. The authors mention briefly at the start of the introduction about high-risk settings where interpretability is needed.

**Main Review:**

*** Originality *** There are many prior works in "interpretable training", and it seems uneasy to name the method after this general field, especially given the prior works. Some works which incorporate interpretability into the training process in a self-supervised fashion include:

- Tell Me Where to Look: Guided Attention Inference Network by Kunpeng Li et. al.
- Sharpen Focus: Learning with Attention Separability and Consistency by Lezi Wang et. al.
- Self-Erasing Network for Integral Object Attention by Qibin Hou et. al. - from NeurIPS 2018
- Object Region Mining with Adversarial Erasing: A Simple Classification to
Semantic Segmentation Approach by Yunchao Wei et. al.

Although "interpretable training" has been around for a while, the authors do not include another interpretable training baseline. And although the papers I have mentioned evaluate on vision tasks, many of the ideas are extendable to the settings which the authors consider here, in the same manner (e.g., running the saliency methods on text instead of images).

*** Quality ***

I did not understand the ability of the box plots to show interpretability. As described in Figure 2: "most features have gradient values around zero with large gaps between mean and outliers, indicating the model’s ability to differentiate between informative and non-informative features." - Why is this true?

Also, I am wary of these results, since they seem to go against Goodhart's Law: "When a measure becomes a target, it ceases to become a good measure." The input gradients were used directly in the model optimization process. If a saliency method is used for evaluating the interpretability, it should be a different one than what was used in optimization - for example, something black-box like RISE (by Vitali Petsiuk et. al.). Furthermore, there is prior work which shows that input gradients are often not interpretable (e.g., Bridging Adversarial Robustness and Gradient Interpretability by Beomsu Kim et. al.).

It is admirable that the authors evaluate their algorithm on many benchmarks, in a variety of forms of data - MNIST, CIFAR, and BIRD for images, Movies, FEVER, and e-SNLI for language, and synthetic time series benchmarks. They also use a variety of model architectures.

*** Clarity *** The paper contains many grammar errors, although they are generally not a hindrance to understanding. For example, Line 119: "Although we use on vanilla gradients during training to mask unimportant features which is not computationally expensive, this training procedure (similar to other training procedures like adversarial training as an example) is more expensive than traditional training."

On Line 138, the statement: "We would expect the background gradient (i.e., most of the features) to be close to zero." This is not always true (e.g., the background can easily be correlated with the object, and thus feature importance on the background is likely to be non-zero). But, the model focusing on object features rather than background features can instead be stated as a desirable quality to have, rather than the expectation.

Typo Line 172 - MINST

*** Significance *** It is great that the method the authors propose is self-supervised, and does not rely on the need for ground truth saliency maps, which are often not available (at least, from true GT annotations. It could be possible to get pseudo-GT saliency guidance from a pretrained model which is, say, good at recognizing "objectness"). The loss function is also nice in its simplicity. However, as stated above, there are many prior works in interpretable training, which this paper has not compared to.

**Time Spent Reviewing:**

2

---

> ### Author Response · Authors · 2021-08-10
> **Response to Reviewer jiTu**
>
> Thank you for your thoughtful review.
>
>
> ***Re: “ There are many prior works in "interpretable training", and it seems uneasy to name the method after this general field, especially given the prior works”*** Thank you for the suggestion. We will consider changing the name of the method to self-supervised interpretable training to be more specific.
>
>
> ***Re: “Although "interpretable training" has been around for a while, the authors do not include another interpretable training baseline.”*** We would like to emphasize that our work is focused on increasing the model interpretability through training (similar to [5,6]) in a self-supervised manner.  We will add papers [1-4] in the related work sections. However, note that work [1,3,4] is from weakly supervised localization literature that uses attention maps to improve segmentation. Sharpen Focus [2] incorporates attention maps into training to improve classification accuracy. Taking Sharpen Focus [2] as an example, we repeated MNIST experiments described in section 5.1 (we choose MNIST rather than other datasets since we can show qualitative results through model accuracy drop; this is explained in the model accuracy drop subsection in lines 149-181). For this experiment, we remove features with high saliency which are identified by the gradient. Features are then replaced by a black background. The accuracy is calculated when removing different percentages of features. Below is the accuracy drop for different models along with the area under the curve (AUC). Note that a larger drop and smaller AUC are better. Not surprisingly, Sharpen focus is even less interpretable than a regular CNN. The reason this is the case is that it has a more complex architecture and is trained to improve accuracy not interpretability. Note that the one common benchmark in papers [1-4] is the PASCAL VOC 2012 image segmentation benchmark, while our paper is focused on interpretability.
>
> |               | 0     | 10   | 20   | 30   | 40   | 50   | 60   | 70   | 80   | 90   | AUC    |
> |---------------|-------|------|------|------|------|------|------|------|------|------|--------|
> | Traditional   | 99.4  | 84.3 | 60.4 | 40.1 | 28.2 | 20.8 | 17.1 | 14.9 | 13.9 | 13.9 | 3360.4 |
> | Interpretable | 99.3  | 49.1 | 20.9 | 11.1 | 8.6  | 8.3  | 8.4  | 9.5  | 10.6 | 11.5 | 1817.6 |
> | Sharpen Focus | 100.0 | 73.4 | 50.0 | 43.7 | 32.8 | 28.1 | 26.6 | 23.4 | 20.3 | 15.6 | 3562.5 |
>
>
> ***Re: “I did not understand the ability of the box plots to show interpretability.”*** The idea is that very small gradients indicate uninformative features while large gradients indicate informative ones [7] (Note by gradients we are referring to input gradients). The proposed training method helps reduce noisy fluctuating gradients in between which is shown in the box plot; we will clarify this in the final paper version.
>
>
> ***Re:”Also, I am wary of these results, since they seem to go against Goodhart's Law:”*** We note that along with input gradients, we have used multiple other saliency methods in our evaluations including Integrated Gradients, DeepLIFT, SmoothGrad, Gradient SHAP and DeepLift with Shapely values, where the masking is done by vanilla gradient and the testing is done by the mentioned methods. Results are shown in figures 3 and 5 in the main paper along with figures 4-7 and tables 5-8 in the supplementary. Our method improves model interpretability even when using other evaluation metrics than the vanilla gradients.
>
> ***Re:”Clarity”*** Thank you for pointing this out. We will make sure to fix all typos and grammatical errors.
>
> ***Re: “On Line 138, the statement: "We would expect the background gradient to be close to zero." This is not always true”*** we will clarify that we ‘desire’ the model to focus on object features rather than background features.
>
>
>
> [1] Tell Me Where to Look: Guided Attention Inference Network by Kunpeng Li et. al.
> [2] Sharpen Focus: Learning with Attention Separability and Consistency by Lezi Wang et. al.
> [3] Self-Erasing Network for Integral Object Attention by Qibin Hou et. al. - from NeurIPS 2018
> [4] Object Region Mining with Adversarial Erasing: A Simple Classification to Semantic Segmentation Approach by Yunchao Wei et. al.
> [5] Reza Ghaeini, Xiaoli Z Fern, Hamed Shahbazi, and Prasad Tadepalli. Saliency learning: Teaching the model where to pay attention.
> [6] Andrew Slavin Ross, Michael C Hughes, and Finale Doshi-Velez. Right for the right reasons: Training differentiable models by constraining their explanations.
> [7] Baehrens, David, et al. "How to explain individual classification decisions.

---

> > ### Comment · Reviewer_jiTu · 2021-08-31
> > **Response to authors**
> >
> > Hi author(s), thanks for the response. I appreciate the clarification that in Figures 3 and 5, the training was still done with vanilla gradients, but the evaluation was a variety of saliency methods. This gives more weight to the results. I will increase my score to a 6. However, I have remaining concerns about the limited comparison to prior work. Thanks for the comparison to Sharpen Focus - it's good to know your proposed method outperforms in terms of accuracy drop and AUC. However, the works [1] and [2] apply to classification as well as segmentation, and do not require external supervision. The "self-erasing" high-level term in [2] could also be used to describe this work. I still see these as direct comparisons. But, I do like the simplicity of the KL divergence loss in this work.
> >
> > I also stand by my first point strongly (which is just around naming the method, so does not really factor into the score) - "interpretable training" is a very broad term, and it doesn't seem right to use it for your specific method - I agree that self-supervised interpretable training is more apt.
> >
> > I agree with other reviewers that the broad applicability of this work is exciting (language, time series, vision).
> >
> > [1] Tell Me Where to Look: Guided Attention Inference Network by Kunpeng Li et. al.
> > [2] Self-Erasing Network for Integral Object Attention by Qibin Hou et. al. - from NeurIPS 2018

---

### Official Review · Reviewer_zFG8 · 2021-07-16

**Rating:** 5
**Confidence:** 3

**Summary:**

The paper proposes a new interpretable training procedure for neural networks. The paper argues that the existing model’s prediction explanation through saliency maps uses backpropagation with noisy gradients. The approaches use gradient calculations to assign an importance score to individual features, reflecting their influence on the model prediction. As the importance scoring is dependent on the Gradients, noisy gradients can result in unfaithful feature attributions.  The problem was attempted to tackle by introducing an interpretable training procedure to reduce noisy gradients of the existing saliency methods: Gradient (GRAD), Integrated Gradients (IG), DeepLIFT (DL), SmoothGrad (SG), and Gradient SHAP (GS). The proposed interpretable training procedure (interpretable training) iteratively masks features with small and potentially noisy gradients while maximizing the similarity of the model outputs for both masked and unmasked inputs using a loss function that combines KL divergence and another loss function.

**Limitations And Societal Impact:**

Yes.

**Main Review:**

The contribution and clarity of the paper can be improved by addressing the following questions:

Re: “most features have gradient values around zero with large gaps between mean and outliers, indicating the model’s ability to differentiate between informative and non-informative features.” --- I don’t see the connection between the large gap in gradient values (between mean and outliers) and models’ ability to differentiate between informative and non-informative features. Gradient values can be related to the long-term dependency in backpropagation algorithms, however, I am not sure how that can be related to informative and non-informative features.

Relevant to the above, Re:” If gradient-based explanation methods faithfully interpret the model’s predictions, irrelevant features should have gradient values close to zero.”: The vanishing gradient problems in neural network (e.g. LSTM) training suggest that the small gradients are related to the longer sequence (long term dependency) due to the repeated application of chain rule in the backpropagation algorithms. Thus, gradients get smaller not because the features in the long sequences are irrelevant.

It would be useful to have elaborated examples presented to show how the small gradients and irrelevant features are related. For example, there are two examples in a relevant previous work [1] show how gradient was used for feature selection.

Re: “S(X) would sort elements of X based on the sum of feature embeddings of each word x ”: What is the interpretation of the sum of the feature embeddings of each word? Feature embeddings are points in a vector space and summing up the coordinate values in that vector space for a word for sorting might not have any meaning.


Minor comments:

Ln 25: “the maps produced are often noisy. ”--- Not sure what does this line means.


[1] Perkins, S., Lacker, K., & Theiler, J. (2003). Grafting: Fast, incremental feature selection by gradient descent in function space. The Journal of Machine Learning Research, 3, 1333-1356.


**Time Spent Reviewing:**

4

---

> ### Author Response · Authors · 2021-08-10
> **Response to Reviewer zFG8**
>
> Thank you for your detailed review.
>
> ***Re: “most features..”***  The gradient of the class score function with respect to the input can be interpreted as a (local) sensitivity map that identifies features that strongly influence the final decision [1]. The derivative of the class function with respect to the input may fluctuate sharply at small scales. This has been addressed in different ways: SmoothGrad[2]  addresses this by adding noise to the input multiple times, calculates the gradient, and takes the average. Our approach addresses this problem by enforcing spares regularization during training.
>
>
> ***Re: "If gradient-based"*** You are correct that in recurrent neural networks there is a vanishing gradient problem for long sequences due to the network forgetting over time. This has been addressed in some architectures like Input-cell-attention [3]. Here we are addressing a different problem where our models don’t suffer from vanishing gradients but there are small fluctuating gradients in inputs. We have benchmarked Input-cell-attention [3] for the synthetic time series dataset and showed that using our proposed training approach we were able to produce improved explanations results (please see Supplementary Table 6.)
>
> ***Re: "It would be useful”*** Thank you for pointing us to the relevant paper. We have demonstrated this somehow empirically in the time series section where synthetic datasets have known informative and non-informative features where through experiments we show that the use of interpretable training improves the precision and recall of the produced explanations (results are in sup. Tables 5-8.) In our updated draft, we will highlight more examples showing the relation between small gradients and irrelevant features.
>
> ***Re: “S(X) would sort elements of X based on the sum of feature embeddings of each word x ”:*** What is the interpretation of the sum of the feature embeddings of each word?” Thank you for pointing this out. Please note that we do not take the sum of the word embedding itself; we take the sum of the saliency of the word embedding as done in standard NLP practice [4]. Since the importance of a word can be represented by the importance of each dimension in the feature embedding.  Sorry for this misunderstanding, we will clarify this in the final draft.
>
> ***Re:“the maps”***  by noisy we are referring to visual noise (similar terminology has been used by [4]). We will clarify this in the final submission.
>
>
> [1] Baehrens, David, et al. "How to explain individual classification decisions."
> [2] Smilkov, Daniel, et al. Smoothgrad: removing noise by adding noise.
> [3] Ismail, Aya Abdelsalam, et al. Input-cell attention reduces vanishing saliency of recurrent neural networks.
> [4] Kokhlikyan, Narine, et al. "Captum: A unified and generic model interpretability library for pytorch."

---

> > ### Comment · Reviewer_zFG8 · 2021-09-01
> > **Re: Response to Reviewer zFG8**
> >
> > Thanks for the responses and clarifications. The notion of explanation in [1] is not related to the prediction error that is used for global feature explanation, but only to the label provided by the prediction algorithm to explain individual inputs. Not sure if informative and non-informative global feature identification based on all the inputs can be interpreted similarly based on gradient values.

---

> > > ### Author Response · Authors · 2021-09-01
> > > **Response to Reviewer zFG8**
> > >
> > > Thank you for your response. Using the gradient of the entire input as a sensitivity map has been done in [1] (section 3.1 Class Saliency Extraction). Then, it was adopted by other works including [2-4].  It has been proven by Ancona et al. [5] that complex gradient-based attribution methods [2-4] can be reformulated as computing backpropagation for a modified gradient function.
> > > The intuition is that for any input, one can construct a sensitivity map by differentiating the class activation function. This sensitivity map represents how much of a difference a tiny change in each feature of the given input would make to the classification score of the class for that input. The resulting map will highlight regions with high influence on the final decision (i.e., informative features).
> > > In our proposed method we look at all features in a single input and use this gradient-based sensitivity map to identify informative/non-informative features. We then incorporate this information to train models that better identify these informative/non-informative features by masking “non-informative features” and minimizing KL divergence between the original input and the masked input. We will add further explanations about this to the revised draft.
> > >
> > > [1] Simonyan, K., Vedaldi, A., & Zisserman, A. (2013). Deep inside convolutional networks: Visualising image classification models and saliency maps.
> > > [2] Smilkov, Daniel, et al. Smoothgrad: removing noise by adding noise.
> > > [3] Sundararajan, Mukund, Ankur Taly, and Qiqi Yan. "Axiomatic attribution for deep networks."
> > > [4] Shrikumar, Avanti, Peyton Greenside, and Anshul Kundaje. "Learning important features through propagating activation differences."
> > > [5] Ancona, Marco, et al. "Towards better understanding of gradient-based attribution methods for deep neural networks."

---

### Decision · Program_Chairs · 2021-09-27

**Decision:**

Accept (Poster)

**Comment:**

The paper proposes a new procedure for training neural networks that achieves improved interpretability by masking the bottom k input gradients, helping address the noise in saliency map techniques. The reviewers found the paper well motivated, liked the experiments (especially the fact that they’re conducted on many domains), and found the method novel.

The reviewers appreciated the thoroughness of the rebuttal, both in terms of experiments and responses, clarifying the points around related works, writing and additional discussions.  I therefore recommend acceptance.   I encourage the authors to integrate all the suggestions from the reviewers into the camera ready.